# A framework for conducting GWAS using repeated measures data with an application to childhood BMI

Kimberley Burrows [1,2,23], Anni Heiskala [3,23], Jonathan P. Bradfield[4,5,23], Zhanna Balkhiyarova[6,7,8,23], Lijiao Ning [9], Mathilde Boissel [9], Yee-Ming Chan [10,11], Philippe Froguel [9,12], Amelie Bonnefond[9,12], Hakon Hakonarson [4], Alexessander Couto Alves [13], Deborah A. Lawlor [1,2], Marika Kaakinen[6,7,14], Marjo-Riitta Järvelin [3,15,16], Struan F. A. Grant [4,17,18,19,20], Kate Tilling [1], Inga Prokopenko [6,7,24], Sylvain Sebert [3,24] ✉, Mickaël Canouil [9,24] & Nicole M. Warrington [21,22,24] ✉

Genetic effects on changes in human traits over time are understudied and may have important pathophysiological impact. We propose a framework that enables data quality control, implements mixed models to evaluate trajectories of change in traits, and estimates phenotypes to identify age-varying genetic effects in GWAS. Using childhood BMI as an example trait, we included 71,336 participants from six cohorts and estimated the slope and area under the BMI curve within four time periods (infancy, early childhood, late childhood and adolescence) for each participant, in addition to the age and BMI at the adiposity peak and the adiposity rebound. GWAS of the 12 estimated phenotypes identified 28 genome-wide significant variants at 13 loci, one of which (in *DAOA*) has not been previously associated with childhood or adult BMI. Genetic studies of changes in human traits over time could uncover unique biological mechanisms influencing quantitative traits.

Genome-wide association studies (GWAS) have been informative over the past two decades in extending our knowledge of the genetic architecture of common complex traits and diseases. The vast majority of GWAS have concentrated on cross-sectional phenotypes (i.e. one measure per person per study). However, many human traits change over time, and there may be a genetic component underlying this dynamic process of change in the trait (see for example[1–7],). Therefore, studying trait trajectories is increasingly important to uncover loci beyond those found from GWAS of cross-sectional traits.

Linear mixed models (LMMs) are often used to assess age- (or time) varying exposure-outcome associations[8]. This statistical model summarises repeated measures data into an average trajectory across the sample (i.e. the fixed effects), as well as the individual variations around this average for each participant within the sample (i.e. the random effects). LMMs can be used to explore age- (time) varying

effects of genetic variants on outcomes. Indeed, there are examples of using LMMs to estimate the association between selected SNPs and trajectories of change in phenotypes (see for example[9–13],). However, scaling this approach up to conduct GWAS on the longitudinal change in a trait can be extremely computationally intensive; for example, analysing ~2.5 million SNPs and longitudinal BMI data in 7916 individuals from the Avon Longitudinal Study of Parents and Children (ALSPAC) took approximately 1440 hours to complete[2], in comparison to a few hours using standard software with a single measure per individual.

There are current statistical methods available to utilise repeated measures of a phenotype within an individual and facilitate the detection of genetic variants that have age- (time) varying effects when the phenotype changes linearly over time. This includes methods that use a two-stage approach, in which (i) a LMM is fit to the repeated

---

measures data and (ii) the best linear unbiased predictors of each individual's trajectory are extracted and used as the outcome in the GWAS[14–16]. These two-stage approaches reduce the computational challenge of fitting the LMM, often allowing the GWAS to be conducted using standard software. As an alternative, Sikorska and colleagues[17] developed a method whereby the variances of the random intercept and slope, along with the variance of the residual, are estimated in a LMM without a genetic variant. This relies on the assumption that the variances do not change when the genetic variant is included in the model. These estimated variances are then used in a large system of linear equations, which is solved to estimate the genetic effects at each SNP. These two-stage methods estimate the association between a genetic variant and a change in a phenotype over time. In contrast, Ko and colleagues (2022) developed a method that estimates the genetic effect on within-subject variability that can be applied to biobank-scale repeated-measures data[18]. All these methods described for longitudinal GWAS analyses are applicable for phenotypes that follow linear trajectories over time[16–18]. However, this is not realistic for many traits, particularly when investigating change in a phenotype over a long period of time.

An example of a trait with a non-linear trajectory is body mass index (BMI) across childhood. The BMI trajectory across childhood starts with a rapid increase soon after birth until the "adiposity peak" (AP) at approximately 9 months of age, followed by a gradual decline until the "adiposity rebound" (AR) around 4–6 years of age, followed by an increase again until the end of puberty and beyond[19]. There is some evidence for age-varying genetic effects on BMI[2,9,20,21]; however, due to the lack of statistical methods for analysing non-linear trajectories on a GWAS scale, relatively little is known about the genetic determinants of the rate of change in BMI across early life.

The aim of this study is to develop a framework to conduct GWAS to detect age- (time) varying genetic effects of non-linear trajectories using standard GWAS software. Due to restrictions in sharing individual participant data, this framework will be applied to individual participant data within each cohort and then GWAS summary statistics will be meta-analysed. The framework comprised of the following four procedures: (1) apply an algorithm to quality control the longitudinal data to ensure that only the most likely outliers are excluded based on within- and between-individual comparisons; (2) specify an appropriate model for a nonlinear growth trajectory using longitudinal data in a range of different cohorts; (3) ensure the chosen model is correctly parameterized; (4) estimate phenotypes that summarise the trajectory for subsequent GWAS analysis. The resulting GWAS summary statistics can be used in downstream analyses, such as genetic correlation and causal modelling. We describe an example standardised protocol via an easy-to-follow R package, called Early Growth Genetics Longitudinal Analysis (EGGLA), to perform each of these steps for BMI across childhood and provide a harmonised, reproducible set of GWAS summary statistics for further downstream analysis.

## Results

Our analysis included participants from six population-based cohorts (Table 1, Supplementary Data 1 and Supplementary Note 2): 1) the Avon Longitudinal Study of Parents and Children (ALSPAC)[22,23], 2) the European subset of Children's Hospital of Philadelphia (CHOP)[24] 3) the African American subset of CHOP[24], 4) the Northern Finland Birth Cohort 1966 (NFBC1966)[25], 5) the Northern Finland Birth Cohort 1986 (NFBC1986)[26], and 6) OBésité de l'Enfant (OBE)[27,28]. We included measures of BMI at multiple times across early life, ranging from two weeks after birth to late adolescence (18 years for all cohorts except OBE, which had data until 16 years).

### Using a published algorithm to clean longitudinal data

We employed a unified approach to data cleaning across cohorts by using the growthcleanr[29] R package, which flags duplicates and

implausible values for exclusion. After data cleaning, we excluded between 4.2% and 16% of BMI measures within each cohort, most of which were excluded due to missing height or weight information or duplicated measures (Supplementary Data 2). The final analysis for the growth modelling comprised 34,818 females (ranging from 308 to 10,814 per cohort) and 36,518 males (ranging from 252 to 12,002 per cohort).

### Developing an R package to model nonlinear growth

We developed an analysis framework to fit nonlinear growth models for GWAS and an associated R package[30,31] (https://m.canouil.dev/eggla/articles/eggla.html) named the EGGLA framework. The EGGLA R package provides four unified protocols to facilitate the analysis framework, including model diagnostics, model selection, running the chosen LMM, and estimating specific phenotypes. The R package was developed to standardise the analysis framework, allowing all six participating cohorts to provide a harmonised, reproducible set of GWAS summary statistics for further downstream analysis. The EGGLA protocols specific to longitudinal modelling of BMI are outlined in Supplementary Fig. 1 and Supplementary Note 3. The EGGLA model diagnostics protocol cycles through each of the three selected LMMs (including a linear spline, cubic spline, or cubic slope functions for age) with various complexities of random effects and correlation structure (no structure and continuous autoregressive correlation structure of order 1 (CAR(1)); see methods for full description of the LMMs. Several reports are output at this stage to inform selection of the most appropriate model to characterise change in the phenotype (BMI in our illustrative application; see Supplementary Data 3 and Supplementary Note 3). To select our preferred model, we compared the model fit of sex-specific analyses in each cohort based on any convergence issues, performance metrics, and visual inspection of the predicted curves.

Several models either failed to converge or presented warning messages within each cohort (more issues were seen in cohorts with higher sample size and a larger number of repeated measures per individual; Table 2 and Supplementary Data 3). In each cohort, at least eight of the sixteen models converged without any issues (Supplementary Data 3). Specifying a CAR(1) correlation structure seemed to cause the most problems with model convergence across the cohorts/sexes, particularly for the models with a cubic slope or linear spline function for age in the fixed effects.

We used Akaike's Information Criterion (AIC) and Bayesian Information Criterion (BIC) performance metrics to define the best-fitting model as they appropriately penalize model complexity. In ALSPAC, CHOP African Americans, females in NFBC1966, and NFBC1986 the best model included a cubic spline function for age in both the fixed and random effects and a CAR(1) correlation structure (Table 2 and Supplementary Data 3). For the males in NFBC1966 and OBE, the best-fitting model included a cubic spline function for age in the fixed effects, a quadratic spline function in the random effects and a CAR(1) correlation structure. Finally, the best fitting model in the male, European American subset of CHOP had a cubic spline function for age in the fixed effects, a cubic slope function in the random effects, and no correlation structure and in the females the best fitting model included a cubic spline function in the fixed effects, linear spline function in the random effects and a CAR(1) correlation structure. Although the cubic spline in both fixed and random effects produced the most favourable performance metrics for most cohorts (Table 2 and Supplementary Data 3), the model fit with warning messages in the CHOP European Americans and males in NFBC1966 suggesting that the next best performing models (that converge without issues for all cohorts and sexes) should be considered. Therefore, our preferred model included a cubic spline function for age in the fixed effects, cubic slope in the random effects and no specified correlation

**Table 1 | Cohort information for each of the cohorts included in the analysis, including summary statistics of the estimated phenotypes**

| Cohort information | ALSPAC | | CHOP - European American | | CHOP - African American | | NFBC1966 | | NFBC1986 | | OBE | |
|---|---|---|---|---|---|---|---|---|---|---|---|---|
| | Male | Female | Male | Female | Male | Female | Male | Female | Male | Female | Male | Female |
| Country | Bristol, UK | | Philadelphia, USA | | Philadelphia, USA | | Northern Finland | | Northern Finland | | France | |
| Ethnicity | European | | European | | African American | | European | | European | | European | |
| Collection type | Pregnancy cohort | | Random collection from Hospital visits | | Random collection from Hospital visits | | Prospective general population-based | | Prospective general population-based | | Retrospective obese children cohort | |
| Year of birth | 1991-1993 | | 1988-present | | 1988-present | | 1966 | | 1985-1986 | | 1981-2001 | |
| Sample size | 7197 | 6818 | 12002 | 10814 | 10533 | 10772 | 3800 | 3280 | 2734 | 2826 | 252 | 308 |
| Number of BMI measures per person | 8.48 (5.41) | 8.83 (5.41) | 13.90 (16.60) | 13.90 (16.40) | 13.40 (12.40) | 12.60 (12.00) | 15.64 (5.67) | 15.90 (5.74) | 19.31 (5.14) | 19.54 (5.32) | 9.59 (2.93) | 9.59 (2.91) |
| Age at adiposity peak (yrs) | 0.77 (0.04) | 0.78 (0.04) | 0.75 (0.04) | 0.77 (0.05) | 0.72 (0.04) | 0.74 (0.05) | 0.73 (0.03) | 0.75 (0.04) | 0.70 (0.03) | 0.73 (0.04) | 0.75 (0.08) | 0.79 (0.06) |
| BMI at adiposity peak (kg/m²) | 17.75 (1.00) | 17.33 (0.99) | 17.58 (0.82) | 17.02 (0.92) | 17.94 (1.02) | 17.46 (1.08) | 18.22 (1.15) | 17.81 (1.17) | 17.75 (1.09) | 17.29 (1.09) | 18.39 (1.03) | 17.83 (1.03) |
| Age at adiposity rebound (yrs) | 5.69 (1.19) | 5.10 (1.34) | 4.90 (1.51) | 4.54 (1.54) | 4.22 (1.20) | 3.83 (1.20) | 5.67 (0.99) | 5.61 (1.01) | 5.06 (1.01) | 4.87 (1.21) | 2.40 (0.67) | 2.12 (0.62) |
| BMI at adiposity rebound (kg/m²) | 15.78 (1.09) | 15.75 (1.17) | 15.78 (1.06) | 15.57 (1.13) | 15.96 (1.27) | 15.82 (1.43) | 15.40 (1.00) | 15.25 (1.16) | 15.59 (1.08) | 15.48 (1.17) | 17.34 (1.40) | 17.28 (1.19) |
| Derived infancy slope (0-0.5 years) | 0.45 (0.02) | 0.47 (0.02) | 0.44 (0.02) | 0.46 (0.02) | 0.46 (0.02) | 0.48 (0.02) | 0.56 (0.02) | 0.58 (0.02) | 0.36 (0.01) | 0.38 (0.01) | 0.54 (0.05) | 0.48 (0.03) |
| Derived childhood slope (1.5-3.5 years) | -0.03 (0.01) | -0.02 (0.01) | -0.03 (0.02) | -0.02 (0.02) | -0.03 (0.02) | -0.02 (0.02) | -0.04 (0.01) | -0.04 (0.01) | -0.03 (0.01) | -0.03 (0.01) | 0.01 (0.02) | 0.01 (0.02) |
| Derived late childhood slope (6.5-10 years) | 0.02 (0.01) | 0.03 (0.01) | 0.03 (0.02) | 0.03 (0.02) | 0.04 (0.02) | 0.05 (0.02) | 0.02 (0.01) | 0.02 (0.01) | 0.03 (0.01) | 0.02 (0.01) | 0.06 (0.01) | 0.06 (0.01) |
| Derived adolescent slope (12-17 years) | 0.03 (0.01) | 0.03 (0.01) | 0.04 (0.02) | 0.04 (0.02) | 0.03 (0.02) | 0.03 (0.02) | 0.03 (0.01) | 0.03 (0.01) | 0.03 (0.02) | 0.03 (0.02) | N/A | N/A |
| Derived infancy AUC (0-0.5 years) | 1.39 (0.03) | 1.38 (0.03) | 1.38 (0.03) | 1.36 (0.03) | 1.39 (0.03) | 1.38 (0.03) | 1.39 (0.03) | 1.37 (0.03) | 1.40 (0.03) | 1.38 (0.03) | 1.40 (0.03) | 1.38 (0.03) |
| Derived childhood AUC (1.5-3.5 years) | 5.65 (0.11) | 5.64 (0.12) | 5.61 (0.11) | 5.57 (0.14) | 5.62 (0.14) | 5.59 (0.16) | 5.61 (0.12) | 5.58 (0.13) | 5.59 (0.12) | 5.56 (0.13) | 5.75 (0.19) | 5.75 (0.16) |
| Derived late childhood AUC (6.5-10 years) | 9.86 (0.32) | 9.94 (0.37) | 9.93 (0.46) | 9.95 (0.50) | 10.14 (0.57) | 10.24 (0.66) | 9.70 (0.28) | 9.69 (0.34) | 9.82 (0.34) | 9.80 (0.39) | 11.19 (0.51) | 11.12 (0.44) |
| Derived adolescent AUC (12-17 years) | 15.07 (0.57) | 15.30 (0.66) | 15.32 (0.96) | 15.43 (0.94) | 15.73 (1.21) | 16.09 (1.28) | 14.76 (0.51) | 14.87 (0.54) | 15.01 (0.65) | 15.01 (0.64) | N/A | N/A |

See Supplementary Data 1 for further information. The summary statistics are presented as mean (standard deviation). The OBE cohort was unable to estimate the adolescent phenotypes due to lack of data after the age of 16 years (N/A for not available).

**Table 2 | Model fit parameters from each cohort for a selection of the linear mixed models**

| Fixed effects | Random effects | Correlation structure | Diagnostic | ALSPAC | | CHOP - European American | | CHOP - African American | | NFBC1966 | | NFBC1986 | | OBE | |
|---|---|---|---|---|---|---|---|---|---|---|---|---|---|---|---|
| | | | | Male | Female | Male | Female | Male | Female | Male | Female | Male | Female | Male | Female |
| Cubic slope | Cubic slope | None | AIC | 221,242 | Warning | 526,192 | 485,080 | 541,869 | 539,112 | 193,736 | 174,305 | 164,482 | 171,941 | 11,200 | 13,108 |
| | | | BIC | 221,386 | | 526,343 | 485,229 | 541,948 | 539,191 | 193,870 | 174,438 | 164,615 | 172,075 | 11,287 | 13,198 |
| Cubic slope | Cubic slope | CAR(1) | AIC | Error | Error | 498,296 | 458,224 | Error | Error | Error | Error | Error | Error | Error | Error |
| | | | BIC | | | 498,457 | 458,383 | | | | | | | | |
| Linear splines | Linear splines | None | AIC | 219,022 | 221,162 | 529,884 | Warning | 510747 | 510,513 | Warning | 171,842 | 161,985 | 169,762 | 11,161 | 13,104 |
| | | | BIC | 219,166 | 221,306 | 530,035 | | 510895 | 510,660 | | 171,975 | 162,118 | 169,896 | 11,248 | 13,194 |
| Linear splines | Linear splines | CAR(1) | AIC | Warning | Warning | Warning | Error | Warning | Warning | 176226 | Warning | 146391 | Error | 10,859 | 12,759 |
| | | | BIC | | | | | | | 176370 | | 146533 | | 10,951 | 12,855 |
| Cubic splines | Cubic splines | None | AIC | 197,967 | 199,863 | Warning | Warning | 498,764 | 494,604 | Warning | 151,067 | 145,187 | 150,469 | 9989 | 11,955 |
| | | | BIC | 198,300 | 200,196 | | | 498,941 | 494,780 | 151,386 | 151,386 | 145,506 | 150,790 | 10,198 | 12,171 |
| Cubic splines | Cubic splines | CAR(1) | AIC | 197,807 | 199,252 | Error | Error | 449,478 | 450,241 | Warning | 145,963 | 134,603 | 140,156 | 9887 | 11,836 |
| | | | BIC | 198,150 | 199,594 | | | 449,665 | 450,427 | | 146,291 | 134,932 | 140,486 | 10,101 | 12,058 |
| Cubic splines | Cubic slope | None | AIC | 199,470 | 201,913 | 525,410 | 486,808 | 499,411 | 493,777 | 169,979 | 152,346 | 146,432 | 156,150 | 10,069 | 12017 |
| | | | BIC | 199,642 | 202,084 | 525,590 | 486,986 | 499,549 | 493,914 | 170,141 | 152,505 | 146,592 | 156,311 | 10,173 | 12,125 |
| Cubic splines | Cubic slope | CAR(1) | AIC | 198,415 | 200,128 | Error | Error | Error | Error | Error | Error | 134,843 | 140,907 | Error | Error |
| | | | BIC | 198,595 | 200,308 | | | | | | | 135,011 | 141,077 | 11,838 | 11,951 |

See Supplementary Data 3 for further information on the model parameters from the other models and for additional details on the specific error and warning messages.

structure. This model, for individual i at age (time) t, is described using Eq. 1:

$$\log_e(\text{BMI}_{it}) = \beta_0 + \sum_{j=1}^{3}\beta_j \text{Age}_{it}^{j} + \sum_{k=1}^{3}\beta_{k+3}(\text{Age}_{it} - \kappa_k)_{+}^{3} + b_{0i}$$
$$+ \sum_{j=1}^{3} b_{ji}\text{Age}_{it}^{j} + \varepsilon_{it} \tag{1}$$

Where the fixed parameters are represented by $\beta_0, \beta_1, \beta_2,...$ and $\kappa_k$ is the k$^{th}$ knot where:

$$(t - \kappa_k)_{+} = \begin{cases} 0 & if\ t \le \kappa_k \\ t - \kappa_k & if\ t > \kappa_k \end{cases} \tag{2}$$

The parameter estimates for the random effects are represented by $b_{0i}, b_{1i}, b_{2i},...$, and $\varepsilon_{it}$ are the error terms (see methods for further description on the model specification).

**Parameterising the chosen model by refining knot placement**

Because our preferred model included a cubic spline function for age, we sought to further optimize the knot placement (i.e. ages at which there is a change in slope) to ensure our model reflected the underlying BMI trajectory appropriately. We started with knot points at two, eight and 12 years based on previous research[2], which modelled BMI data from one to 17 years of age. However, our modelling of slightly younger ages could require different knot points. To this end, we ran models incrementing the first-knot point by half-year increments and second-knot point by yearly increments; we decided to not move the third-knot point as there is less change in the BMI trajectory after the adiposity rebound (i.e. by 12 years of age) and both our study and Warrington et al.[2] included data across this time period (i.e. 12-17 years). In addition to the previous metrics used for selecting a preferred model, we estimated the age and BMI at the AP and AR to help us distinguish between models (see methods for description on how the AP and AR were estimated).

Previous studies exploring the age of AP in European populations report an average age at approximately nine months[32]. In all the cohorts in this study, later placement of the first knot (at age one, one and a half, or two years) resulted in an increase in the estimated average age at AP (Supplementary Fig. 2). The average age at AP when applying the first knot at one year of age across all included cohorts was 0.75 years (standard deviation (SD) 0.05; Table 1 and Supplementary Data 1), whereas it was over one year of age when the first knot was at two years of age. In addition, the performance metrics were improved in all the cohorts when applying the first knot at one year of age compared to at two years of age (Supplementary Fig. 2). We therefore chose to apply the first knot at one year of age for our final preferred model.

When comparing the model fit while moving the second knot (i.e. testing a knot at age six, seven, or eight years) we found there was very little variation in the performance metrics or in the estimated average age at AP and AR across the cohorts (Supplementary Fig. 2). Therefore, we chose to keep knot two at our a priori age of eight years.

Figure 1 shows the average BMI trajectories predicted from the fixed effects of our chosen model (with knot points at one, eight and 12 years) for each of the six cohorts. The OBE cohort had a substantially higher average BMI and steeper trajectory throughout early life, reflecting the obese children recruited into the cohort (the BMI trajectory in OBE is from two weeks to 16 years of age as there were very few observations beyond age 16 years). The African American subset of the CHOP cohort had an earlier AR and steeper BMI trajectory throughout childhood than the other cohorts. The model fit the BMI data well in all cohorts tested (Supplementary Fig. 3).

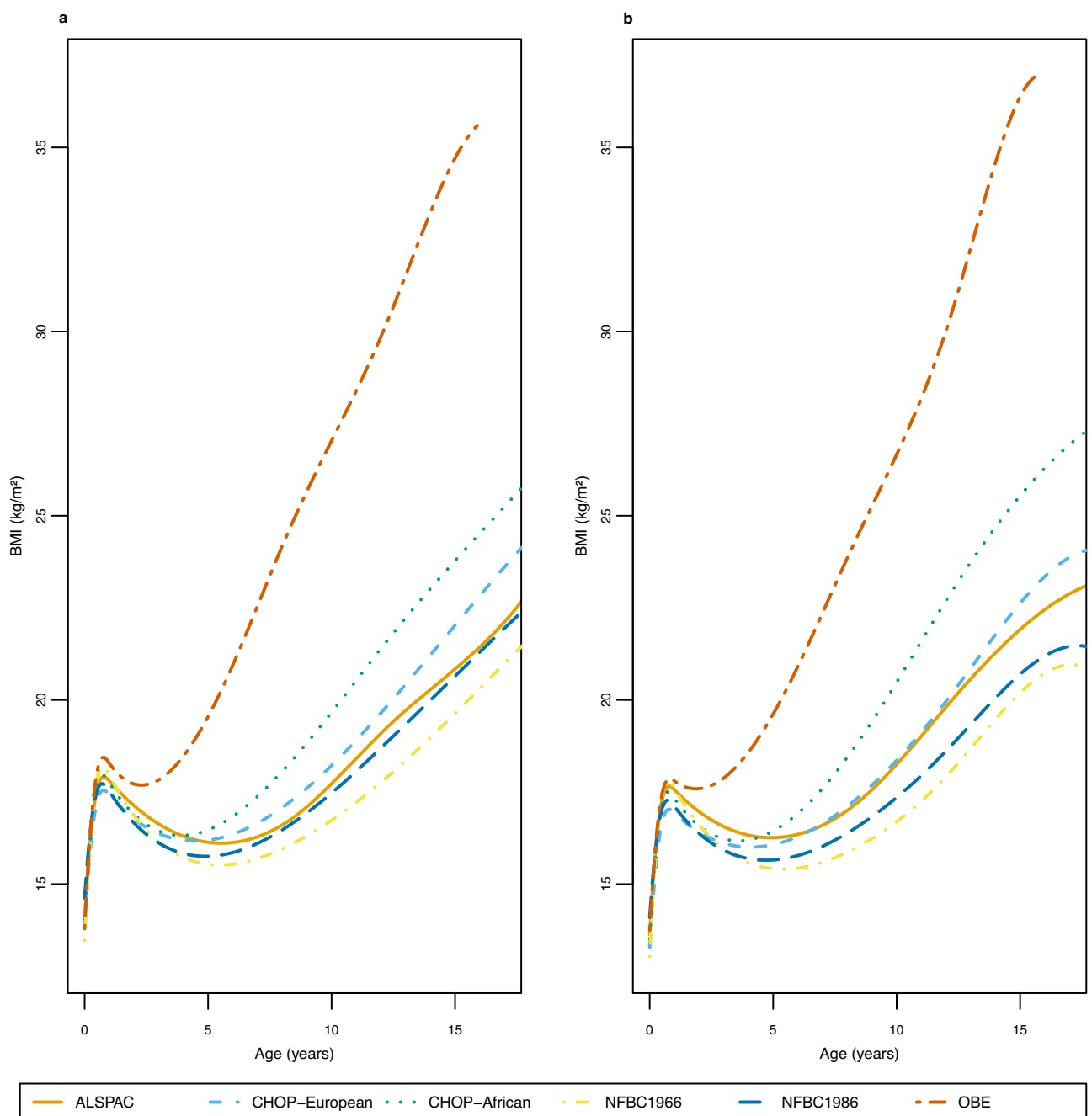

**Fig. 1 | Average BMI trajectories predicted by our final chosen model (cubic spline function in the fixed effects with cubic slope function in the random effects) for each of the six cohorts.** Males are presented in panel (**a**) and females in panel (**b**). BMI trajectories were predicted from 2 weeks to 17 years, which corresponds to the age range that the slopes and AUCs were predicted from, in all cohorts except OBE where they were predicted from 2 weeks to 16 years due to the lack of data after age 16. The year(s) of recruitment for each cohort are as follows: ALSPAC: 1991–1993, CHOP: 1988-present, NFBC1966: 1966, NFBC1986: 1985–1986, OBE: 1981–2001. Source data are provided as a Source Data file.

### Estimating phenotypes that summarise the BMI trajectory

To estimate phenotypes from the BMI trajectory for GWAS analysis, we defined intervals of approximately linear change in BMI. We defined four-time windows: infancy (two weeks to six months), early childhood (1.5–3.5 years), late childhood (6.5–10 years) and adolescence (12–17 years). We calculated a BMI trajectory for each individual within each cohort by combining the estimated fixed effects, which are shared by all subjects within each sex within a cohort, with the predicted random effects, which are specific to each individual. We subsequently estimated phenotypes within each time window from the individual-specific BMI trajectories, including slopes and area under the BMI

curve (AUC), in addition to age and BMI at the AP and AR. We did not estimate the adolescent phenotypes (slope and AUC) in OBE as they only had data to age 16 years.

A summary of each of the estimated phenotypes is presented in Table 1 and Supplementary Data 1. The estimated phenotypes illustrated the expected differences between the cohorts, indicating that our methods are generalizable to a range of cohorts. For example, individuals in the NFBC1966 and OBE cohorts had a steeper infancy slope (0.56 log BMI units per year for males in NFBC1966 and 0.54 log BMI units per year in OBE), in contrast to individuals in NFBC1986 (0.36 log BMI units per year in males). This is equivalent to a 75%

change in BMI over the first year for males in NFBC1966, which equates to approximately 10 kg/m² over the first year if BMI is 14 kg/m² at two weeks of age (approximate BMI at two weeks of age across the cohorts based on the height and weight data in Supplementary Data 1). Whereas in NFBC1986 males, it is equivalent to a 43% change in BMI, equating to approximately 6 kg/m² over the first year if BMI is 14 kg/m² at two weeks of age. This was also reflected in NFBC1966 and OBE having a higher BMI at the AP (for example, 18.22 kg/m² in males from NFBC1966 and 18.39 kg/m² from OBE) while NFBC1986 had a lower BMI at AP (17.75 kg/m² in males). As seen in Fig. 1 and Table 1, the age at AR was earlier for OBE (for example, 2.40 years for males) than the other cohorts (average age of 3.83-5.67 years). The mean rate of growth was similar across the cohorts during late childhood and adolescence, with the mean growth rate ranging between 0.02-0.06 log BMI units per year (Table 1 and Supplementary Data 1), which is equivalent to a 2–6% change in BMI per year.

We investigated the correlation between each of the estimated phenotypes (Supplementary Data 4) and found a high correlation (r > 0.70) between several phenotypes consistently across the cohorts. For example, the infant and early childhood slopes showed a positive correlation (r ≥ 0.78) in both males and females in all cohorts. Both the infant (r ≥ 0.86) and early childhood (r ≥ 0.73) slopes were positively correlated with the age at AP. The late childhood slope was negatively correlated with the age at AR (r ≤ −0.84), which indicates that individuals with an earlier AR have a steeper slope from 6.5 to ten years. These correlations differed in the OBE cohort that included only children with obesity; the correlation between late childhood slope and age at AR were small for males (r = −0.09) and females (r = −0.10). The AUC's generally showed strong positive correlations with the BMI at the AP and AR (infant AUC with BMI at AP, child AUC with BMI at both AP and AR and late child and adolescent AUC with BMI at AR). None of the estimated phenotypes were strongly correlated with the adolescent slope in any cohort (all r < 0.7).

We performed sensitivity analyses in the ALSPAC and NFBC1986 cohorts where we changed the random effects and the correlation structure in the LMM and used these updated models to re-estimate the phenotypes within each time window. The estimated phenotypes were relatively robust to these changes in the underlying LMM (Supplementary Fig. 4 and Supplementary Fig. 5) and the correlations between the estimated phenotypes were similar (Supplementary Data 5).

Finally, all of the estimated phenotypes were associated with BMI at age 18 years (16 years in OBE, as BMI at 18 years was not available), and the strength of the association increased over age (Supplementary Fig. 6). For example, the adolescent AUC explained the majority of the variance in BMI at 18 years (between 74–91% across the cohorts), in contrast to a small amount of variance explained by the infancy AUC (between 0 and 10%)

## Meta-analysing the GWAS summary statistics

We conducted GWAS for each of the estimated phenotypes within each cohort, then combined the results from the GWAS in cohorts with individuals of European ancestry using fixed effects inverse-variance meta-analysis (combined sample size N = 19,308; OBE was excluded from the meta-analysis of the adolescent phenotypes as they did not have data beyond age 16 years). There was no evidence of heterogeneity across the genome in our fixed-effects meta-analysis for the majority of our estimated phenotypes (Supplementary Fig. 7). There was some inflation observed for the early childhood slope and the late childhood AUC, but this was driven by the inclusion of the OBE cohort (see Supplementary Fig. 8 for Q-Q plots excluding OBE). Using the summary statistics from the meta-analysis, we estimated the SNP-based heritability of each of the estimated phenotypes and the genetic correlation between phenotypes using linkage disequilibrium score regression (LDSC). The SNP-based heritability of the estimated phenotypes ranged from 6–28% (Fig. 2), which is comparable to the SNP-

based heritability of BMI across childhood (15–45%[33]) and adulthood (22%[34]). Estimates of SNP-based heritability for the infancy slope, adolescent slope and age at AP were less than 10%, with large standard errors, indicating that the genetic correlation estimates with these phenotypes may not be reliable; however, we have presented them for completeness. AUC in infancy was genetically correlated with AUC in early childhood ($r_g = 0.91$) and the BMI at both AP ($r_g = 1$) and AR ($r_g = 0.66$) but showed relatively low genetic correlations with all the other estimated phenotypes ($r_g < 0.43$), indicating a unique genetic profile for BMI during infancy. The genetic correlation between AUCs in subsequent time periods were high; for example, the genetic correlation between AUC in infancy and early childhood was $r_g = 0.91$, AUC in early childhood and late childhood $r_g = 0.79$, and AUC in late childhood and adolescence $r_g = 0.96$. This indicates that there could be a common set of genes related to BMI across time.

Using LDSC and publicly available summary statistics, we estimated the genetic correlation between our estimated phenotypes based on longitudinal data and childhood BMI at different ages[33] and adult BMI using cross-sectional data[34]. We found a high genetic correlation between our estimated AUCs and childhood BMI measured cross-sectionally at the beginning and end of each age window (Supplementary Data 6). For example, the genetic correlation between early childhood AUC and BMI at age 1.5 years (the beginning of our early childhood time window) was 0.80 (SE = 0.08), which was similar to the genetic correlation of 0.82 (SE = 0.09) with BMI at age 3 years (near the end of our early childhood time window). In contrast, we estimated a moderate genetic correlation between our estimated slopes and childhood BMI (Supplementary Data 6), where the genetic correlation between early childhood slope and BMI at age 1.5 and 3 years of age was 0.13 (SE = 0.09) and 0.39 (SE = 0.10) respectively.

We identified 28 genome-wide significant ($P < 5 \times 10^{-8}$) variants at 13 loci associated with at least one of the 12 estimated phenotypes (Figs. 3–5, Supplementary Data 7). The number of estimated phenotypes the loci were associated with ranged from one phenotype (*LEPR* associated with only age at AR and *DAOA* associated with only early childhood AUC) to seven phenotypes (*SEC16B* was associated with age and BMI at AR, age at AP, late childhood and adolescent AUC, and early and late childhood slope). Of these 13 loci, 12 have previously been identified in GWAS of adulthood BMI or obesity-related traits and nine have been associated with childhood BMI-related traits. We identified one locus in the *DAOA* region on chromosome 13 that was most strongly associated with AUC in early childhood (1.5–3.5 years). Here, the A allele at rs79577162 (*DAOA*) reduces the AUC within this time-period (effect size = −0.02 log BMI years, $P = 4 \times 10^{-8}$). Across the other estimated phenotypes, the A allele at rs79577162 also decreases BMI at the AP (effect size = −0.13 kg/m², $P = 2 \times 10^{-6}$) and AR (effect size = −0.14 kg/m², $P = 5 \times 10^{-6}$), and decreases the AUC across all time periods (effect size for infancy AUC = −0.003 log BMI years, $P = 3 \times 10^{-5}$; effect size for late childhood AUC = −0.04 log BMI years, $P = 1 \times 10^{-4}$; effect size for adolescent AUC = −0.06 log BMI years, $P = 3 \times 10^{-3}$); there was no evidence that the SNP impacts the slope across childhood (all $P > 0.24$) or the age at the AP ($P = 0.26$) or AR ($P = 0.25$). There was some evidence of heterogeneity ($P < 0.05$) at variants in the *SEC16B*, *ADCY3*, *OLFM4* and *FTO* loci (Supplementary Data 8). Results were similar when OBE was excluded from the meta-analysis (Supplementary Fig. 9). Twelve of the 28 the genome-wide significant variants identified in the European meta-analysis showed the same direction of effect in the CHOP African American subset ($N = 6332$) and reached nominal significance ($P < 0.05$) for at least one estimated phenotype (Supplementary Data 8), and a further 12 loci showed the same direction of effect ($P > 0.05$).

The adolescent slope was not strongly associated with any region of the genome (Fig. 4). The most significant locus was near *FAM120AOS* (rs11790060, $P = 7 \times 10^{-8}$) on chromosome 9, a region previously associated with change in BMI over early life[2] (Supplementary Fig. 10).

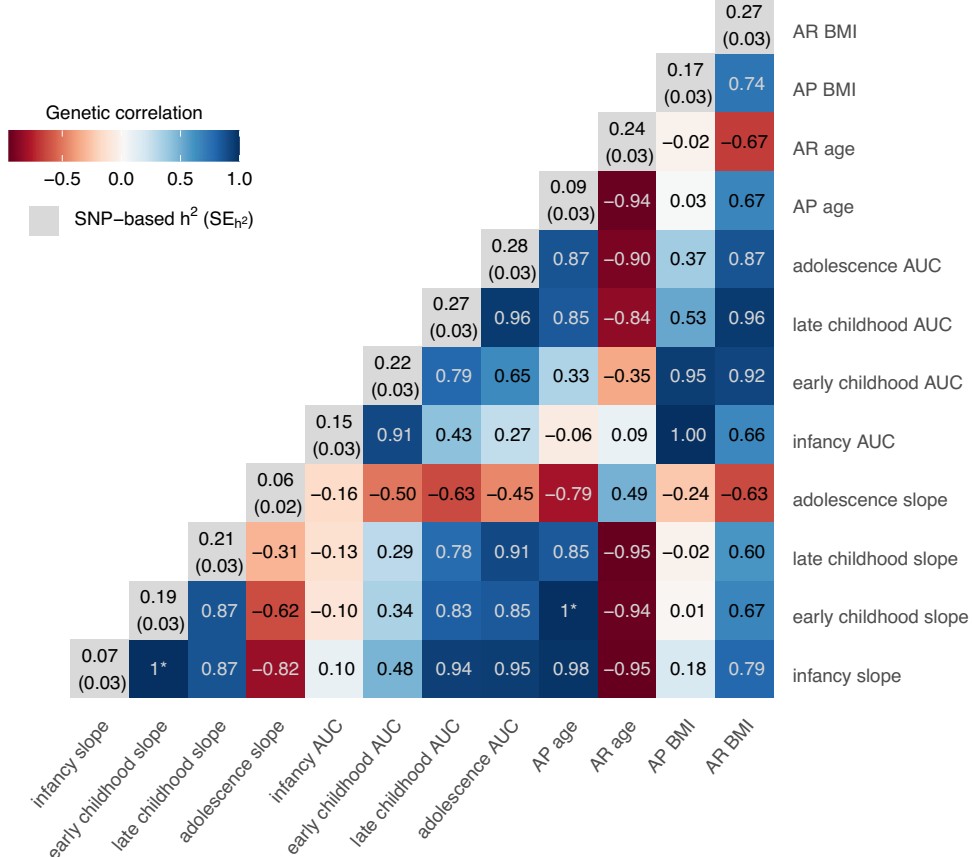

**Fig. 2 | SNP-based heritability (h²) and genome-wide genetic correlation (r$_g$) between the twelve estimated phenotypes summarising growth across early life.** SNP-based heritabilities, presented on the diagonals with standard errors in brackets (SE$_{h²}$), and genetic correlations, presented on the off diagonals, were derived using linkage disequilibrium score regression. SNP-based heritabilities for the age at the adiposity peak (AP age), infancy and adolescent slope are low, with high standard errors (resulting in a z-score <4), and therefore the genetic correlations with these traits are unreliable but are shown for completeness. *estimates of genetic correlation were >1; given this is not possible we have set these to one. AP=adiposity peak, AR=adiposity rebound, AUC=area under the curve. Source data for the genetic correlations are provided as a Source Data file. Source data for the heritability estimates are available in Supplementary Data 6.

Variants near *SEC16B*, which have previously been shown to associate with adult BMI[35], were associated with the majority of our estimated phenotypes. For example, the T allele at rs509325 is associated with decreased age at the AP (effect size = −0.003 years, $P = 1 \times 10^{-8}$), and it is also associated with increased age at the AR (effect size=0.12 years, $P = 5 \times 10^{-16}$) and decreased BMI at the AR (effect size = −0.08 kg/m², $P = 5 \times 10^{-8}$). The T allele is also associated with decreased early childhood slope between 1.5 and 3.5 years (-0.0011 log BMI units per year, $P = 2 \times 10^{-11}$) and late childhood slope between 6.5 and 10 years (-0.0013 log BMI units per year, $P = 1 \times 10^{-17}$), resulting in a lower AUC in the subsequent time periods (i.e. -0.0315 log BMI years, $P = 3 \times 10^{-13}$ for late childhood AUC and -0.0670 log BMI years, $P = 4 \times 10^{-17}$ for adolescent AUC). Variants near *FTO* and *ADCY3* are similarly associated with a number of the estimated phenotypes.

We also investigated the association between 112 unique SNPs previously identified for childhood BMI or obesity-associated traits and our estimated phenotypes (Supplementary Data 9). The majority of the 112 SNPs showed directionally concordant effects between the published BMI traits and our estimated phenotypes (76/112 SNPs increased infancy slope, 86/112 increased early childhood slope, 84/112 increased late childhood slope, 83/112 increased infancy AUC, 97/112 increased early childhood AUC, 96/112 increased late childhood AUC, 96/112 increased adolescent AUC, 80/112 increased age at AP, 89/112 increased BMI at AP, 100/112 increased BMI at AR), with the exception of the adolescent slope (24/112). Given that an earlier age at AR is associated with higher BMI in later life, 82/112 SNPs associated with higher BMI traits were associated with earlier age at AR. At least half of the SNPs reached nominal significance (P < 0.05) for the childhood to adolescent traits, except for the adolescent slope, for which only 18/112 SNPs reached nominal significance.

## Discussion

We have developed a framework to perform non-linear growth modelling and conduct GWAS to detect age- (time) varying genetic effects of non-linear trajectories. To perform the analysis for BMI across childhood in multiple cohorts using our framework, we created an easy-to-use R package, called EGGLA. We consider childhood BMI an important and valuable example for testing our framework due to the established complex changes in adiposity that occur during the childhood, such as the AR and pubertal effects on BMI. We have shown that a LMM with a cubic spline function in the fixed effects and a cubic slope in the random effects fit well in all the cohorts tested, which is similar to a model used previously[2]. We subsequently estimated phenotypes from this LMM and used them to conduct GWAS analyses (N = 19,308 individuals of European descent in the meta-analysis and N = 6332 individuals of African American descent). We identified 28 SNPs from 13 loci that associate with one or more of the estimated childhood BMI phenotypes, one of which has not been previously associated with childhood or adult BMI. These include six loci that were associated with the estimated AUCs, six loci that were associated with both change in BMI over time (estimated slopes) and the estimated AUCs and one that was associated with the age at AR but none

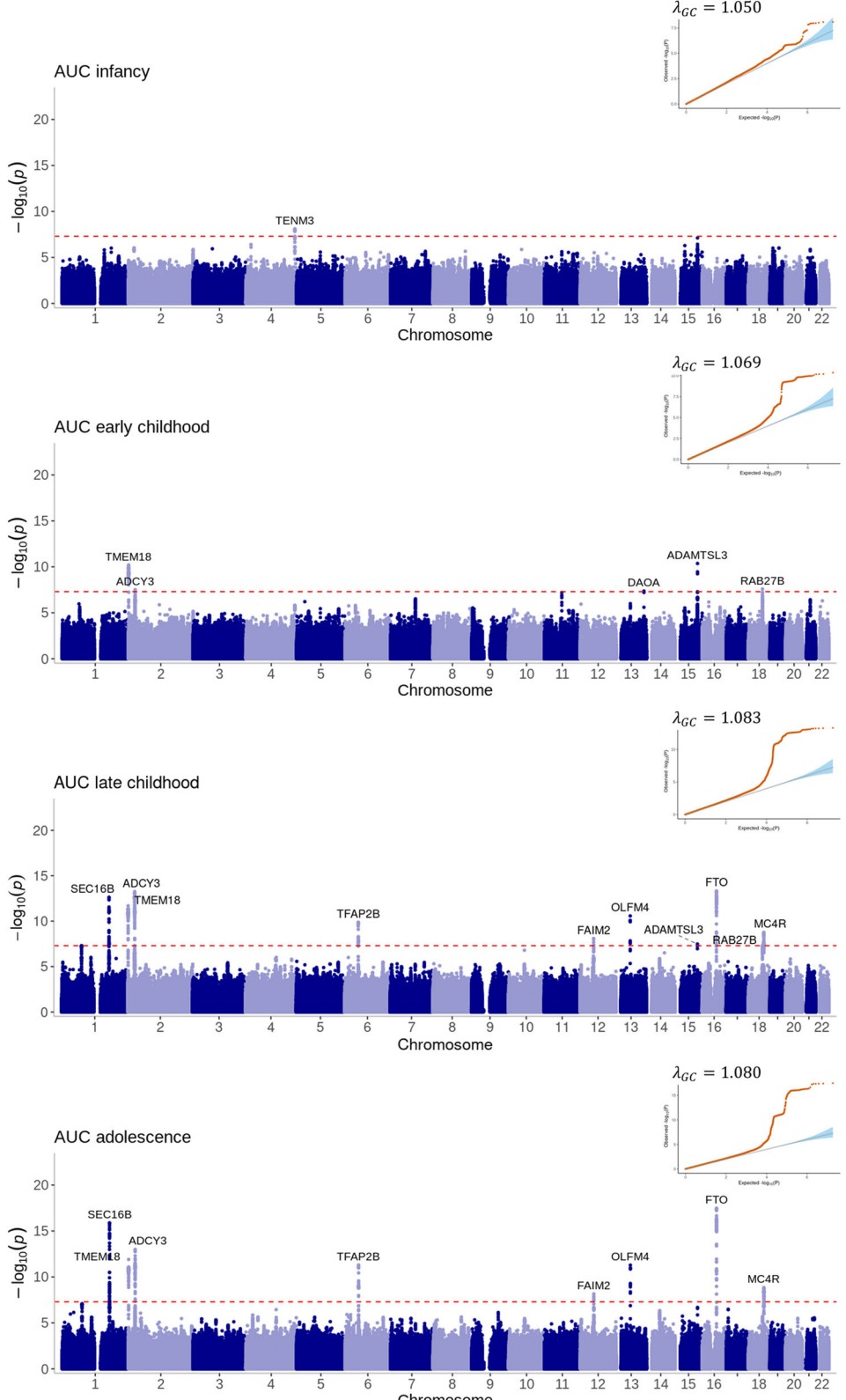

**Fig. 3 | Manhattan plots and quantile-quantile (QQ) plots of the meta-analyses for the area under the curve estimated phenotypes across infancy (0–0.5 years), early childhood (1.5–3.5 years), late childhood (6.5–10 years) and adolescence (12–17 years).** The two-sided association *P* value on the −log₁₀ scale obtained from the inverse-variance-weighted fixed-effects meta-analysis for each of the SNPs (y-axis) was plotted against the genomic position (NCBI Build 37; x-axis). Loci are labelled with their nearest gene annotated by LocusZoom. The red dotted line in the Manhattan plots corresponds to the genome-wide significance level of $P < 5 \times 10^{-8}$, which accounts for multiple testing. The red dots in the QQ plots are the two-sided association P-values, the blue shading represents the 95% confidence bands of the expected values. $\lambda_{gc}$ is the genomic inflation factor.

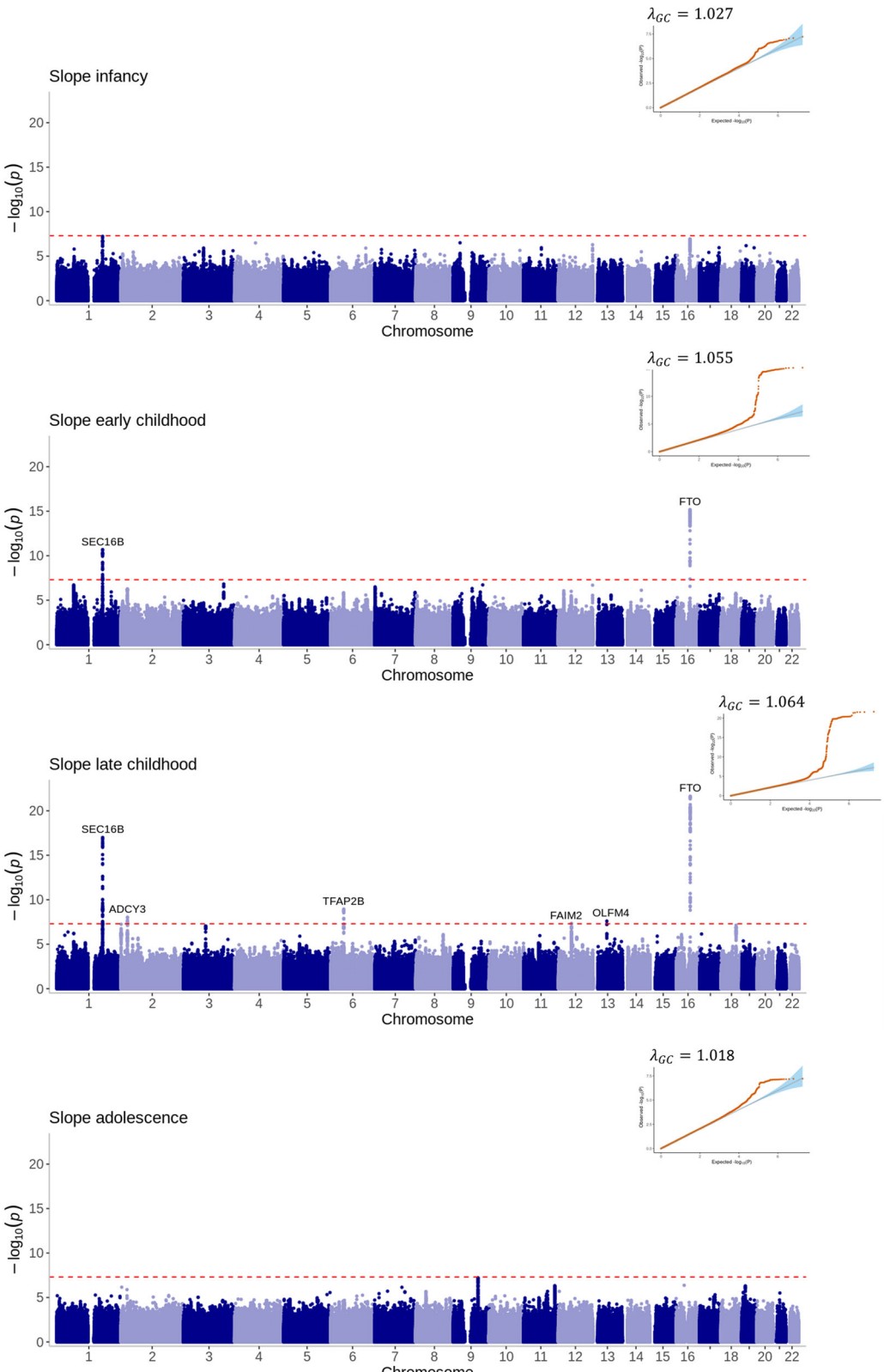

**Fig. 4 | Manhattan plots and quantile-quantile (QQ) plots of the meta-analyses for the slope estimated phenotypes across infancy (0–0.5 years), early childhood (1.5–3.5 years), late childhood (6.5–10 years) and adolescence (12–17 years).** The two-sided association $P$ value on the $-\log_{10}$ scale obtained from the inverse-variance-weighted fixed-effects meta-analysis for each of the SNPs (y-axis) was plotted against the genomic position (NCBI Build 37; x-axis). Loci are labelled with their nearest gene annotated by LocusZoom. The red dotted line in the Manhattan plots corresponds to the genome-wide significance level of $P < 5 \times 10^{-8}$, which accounts for multiple testing. The red dots in the QQ plots are the two-sided association $P$ values, the blue shading represents the 95% confidence bands of the expected values. $\lambda_{gc}$ is the genomic inflation factor.

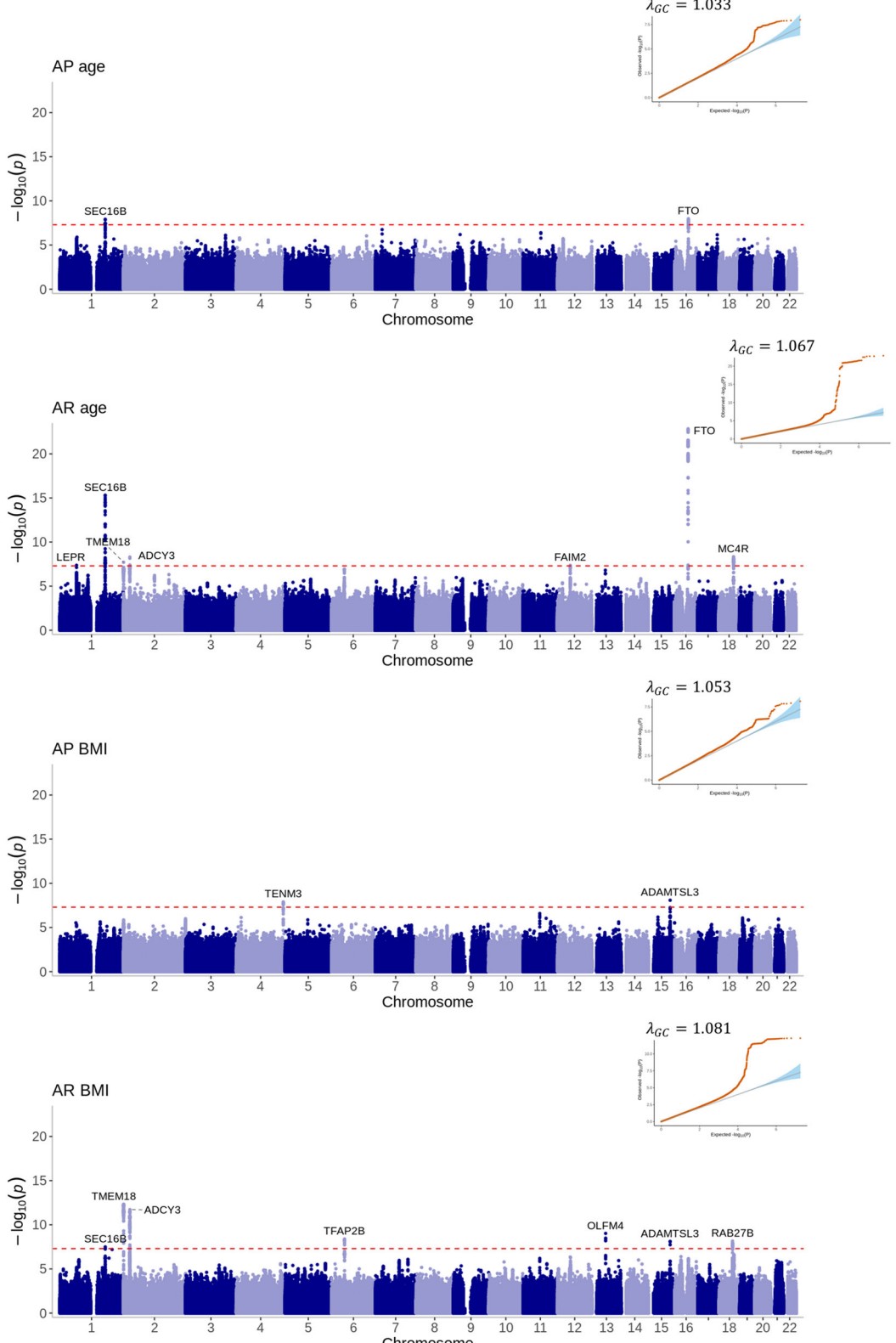

**Fig. 5 | Manhattan plots and quantile-quantile (QQ) plots of the meta-analyses for the age and BMI at adiposity peak and adiposity rebound estimated phenotypes.** The two-sided association P-value on the $-\log_{10}$ scale obtained from the inverse-variance-weighted fixed-effects meta-analysis for each of the SNPs (y-axis) was plotted against the genomic position (NCBI Build 37; x-axis). Loci are labelled with their nearest gene annotated by LocusZoom. The red dotted line in the Manhattan plots corresponds to the genome-wide significance level of $P < 5\times10^{-8}$, which accounts for multiple testing. The red dots in the QQ plots are the two-sided association P-values, the blue shading represents the 95% confidence bands of the expected values. $\lambda_{gc}$ is the genomic inflation factor.

of the AUCs or slopes. Although the majority of the identified loci had previously been identified in GWAS of BMI measured cross-sectionally in childhood and/or adulthood, our framework allows exploration of how the genetic effect changes over age (time), which is difficult to elucidate from cross-sectional GWAS analyses. With a larger sample size, this framework is likely to be useful to explore GWAS of time-varying phenotypes to identify genetic associations that are relatively stable across large age periods (i.e. through the association with estimated AUCs) and those that vary with age (through the association with estimated slopes).

Given we have demonstrated that our framework works well with the complex changes in BMI across childhood, we believe it is generalizable to longitudinal analysis of other traits. For example, the same procedure could be applied to height trajectories across childhood or changes in lean and fat mass. It could also be applied to other non-anthropometric traits, such as blood pressure or cholesterol trajectories across adulthood. The challenging aspect that will require further development for each trait of interest is defining an appropriate model to fit the repeated measures data that accurately describes the trajectory. However, once an appropriate model has been identified for the particular trait, then phenotypes can be estimated for subsequent GWAS analyses. We have provided the R package appropriate for modelling BMI trajectories across early life and details of the framework here so that it can be explored for use with other traits.

Our model fit the BMI data across a range of cohorts, including different ethnicities and different time points for data collection. For example, the final model had a similar fit in the two subsets of the CHOP cohort (the European American and African American), and the average BMI trajectories predicted from the model followed the expected growth patterns from previous research[36], with the African American cohort having a higher BMI and faster rate of growth than the European American cohort. In the OBE cohort, there was very sparse data between the ages of 16 and 18 years; when we removed data after 16 years from the modelling, the model fit in OBE was comparable to the other cohorts. It is unknown how these models will perform in cohorts with data across a shorter age range; it may be necessary to incorporate alternative methods when cohorts are included with data across different age ranges.

We acknowledge that we experienced several convergence issues in the range of models we applied across the cohorts. Although it is not ideal to select models based on whether they converge, it is a practical solution when attempting to apply the same model to a range of datasets with different data structures. Some practical advice to others experiencing convergence issues when attempting to implement our framework include centring the age variable (particularly when zero is not within the age range as the model can struggle to extrapolate to zero), ensuring that each cohort has enough repeated measures of the phenotype (more repeated measurements per individual will allow a more complex random effects structure to be fit), testing different optimization algorithms and, finally, if one of several cohorts is particularly problematic in terms of convergence then investigate the data structure of that cohort and assess whether it is important to include them in the analyses.

We propose using the AUC as one of the summary measures of the trait trajectories. The AUC for childhood BMI has been described as "the child's cumulative 'exposure' to excessive body weight"[37], and has been used in epidemiological studies. Using AUC has potential benefits over analysing BMI in a cross-sectional manner as it is a combination of both baseline BMI and the incremental change in BMI over the time-period, whereas cross-sectional BMI would only capture the BMI at a given time point. Additionally, using the AUC rather than BMI at a single time point potentially increases statistical power to detect a genetic effect as the multiple BMI measurements used to estimate the AUC would average out any errors in the measurements and therefore reduce the variance attributable to measurement error. The AUC was more highly phenotypically and genetically correlated with BMI at the AP and AR, as well as at age 18, whereas the slopes were more highly correlated with the timing of the AP and AR. Consistent with the correlations, the SNP effects for the AUC were more directionally concordant with previously identified BMI and obesity loci than the SNP effects for the slope. This indicates, at least for BMI in early life, that genetic studies of the estimated slope parameters could uncover biological mechanisms driving BMI in childhood that the current studies of childhood BMI have failed to identify. In contrast, genetic studies of the estimated AUC parameters would be likely to provide similar findings to the current genetic studies of childhood BMI; however, because the AUC parameters are estimated from the trajectory, measurement of BMI at the exact same time points across cohorts is not necessary, so using AUC allows incorporation of more cohorts to enhance sample sizes and statistical power for genetic studies.

Our analyses of BMI showed consistent results with previous literature, further validating our approach. First, the phenotypic correlations between the age and BMI at the AP and AR are similar to the correlations from a paediatric cohort with data collected between 1980–2008[37]. For example, Wen et al.[37] estimated the correlation between BMI at the AP and AR to be 0.76, and ours ranged between 0.69–0.91 across the cohorts. Similarly, their correlation between age and BMI at the AR was -0.48, and ours ranged between -0.39 to -0.67. This indicates that the dependencies between the estimated phenotypes are consistent regardless of the underlying LMM fit to the repeated measures data, which is consistent with our comparison of estimated phenotypes using different structures to model within-individual variation. Second, the genetic correlations between the infancy AUC/early childhood AUC/BMI at AP and the other estimated phenotypes were relatively low, indicating that genetic loci associated with BMI in the first 3–4 years of life are likely to be different from those associated with BMI in later life. This observation is consistent with both a twin study that has shown that the genetic correlations differ between early and middle childhood[38] and the MoBa study that show that the genetic correlation between childhood and adult BMI dramatically increases after the age of 5 years[33]. Third, the SNP-based heritability of the estimated phenotypes ranged from 6–28% and is similar to the SNP-based heritability of cross-sectional BMI in childhood (15–45%[33],) and adulthood (22%[34],). The low SNP-based heritability observed for the infancy slope (0.07 (SD = 0.03)) and age at AP (0.09 (SD = 0.03)) was also seen by Couto Alves and colleagues[39] (-0.03 (SD = 0.08) for age AP). This low SNP-based heritability could be due to a higher environmental component operating at this age, a genetic component that is not tagged by the SNPs on the GWAS array or a maternal genetic component that is independent of the child's genetic component. Alternatively, given the SNP-based heritability from cross-sectional BMI GWAS between 6 weeks of age and 1 year in the MoBa study ranged from 0.2–0.45[33], it could indicate that our estimated phenotypes are not proxying the underlying BMI trajectory well through this age range. Therefore, further investigation into this low SNP-based heritability is warranted.

The effect estimates at identified SNPs in our GWAS are consistent with previously reported effects on childhood growth. For example, Warrington et al.[2] identified rs1558902 at the *FTO* locus to be associated with change in BMI over childhood, which is in high LD with rs55872725 (D' = 1, r2 = 1)[2]. We found that the T allele at rs55872725 is associated with increased rate of BMI change from infancy to late childhood ($P < 7 \times 10^{-7}$), but not associated with adolescent slope ($P = 0.129$), which is consistent with the pattern of association identified by Warrington et al.[2] Additionally, the T allele is associated with lower AUC in infancy ($P = 0.002$), not associated with AUC in early childhood ($P = 0.502$) and then associated with higher AUC in late childhood and adolescence ($P < 4 \times 10^{-14}$), which again is consistent with Warrington et al[2]. where the genetic effect is associated with

decreased BMI from 1–2 years of age, is not associated with BMI from 3 to 5 years of age, then is associated with increased BMI from 6 years of age onwards[2]. Therefore, although we have used different methods to identify and describe the genetic effects on growth, we are able to recapitulate what has previously been described, validating our proposed framework.

We have used our model to estimate the BMI and age at both the AP and AR. However, we acknowledge that the biological significance of these two markers remains unclear. It was initially thought that the relationship between age at AR and later risk of obesity was due to both the number and size of adipocytes increasing[19]. Later studies propose that it could be related to an increase in the rate of lean mass rather than fat mass development[40,41]. Cole[42] suggests that it is a statistical phenomenon driven by both high centile and upward centile crossing, which are separately associated with an early rebound. Our findings here, and further application of our framework to change in fat and lean mass across childhood, could provide insights related to these different hypotheses.

There are several limitations to our approach. Firstly, the trajectories for each individual (i.e. the best linear unbiased predictors) are biased towards the average trajectory of the cohort (a property of LMMs known as 'shrinkage'), particularly when the individual has fewer repeated measures or when their trajectory is vastly different from others in the cohort[43]. This could result in a biased estimate of the SNP effect on the trajectory in GWAS. Future work could incorporate approaches to adjust for the bias introduced by shrinkage[14–16] into our framework. Secondly, we have not accounted for the high phenotypic and genetic correlation between the estimated phenotypes (see Fig. 2 and Supplementary Data 4) in the GWAS meta-analysis. Performing a multivariate meta-analysis accounting for the high correlation between the estimated phenotypes may reduce the overlap between the genetic signals seen. It may also increase the power to detect a genetic locus by leveraging information from the other correlated phenotypes, as seen in analyses using the MTAG software[44]. Further research on the most appropriate multivariate meta-analysis method is required. Third, we have only tested our framework on relatively small cohorts with large numbers of repeated measurements and it is unclear how this will scale to biobank-size studies with over one hundred thousand individuals. For instance, in ALSPAC where there are 6,818 samples and 60,169 observations within females, the compute time for each model ranged from 0.04 minutes for the model with a cubic slope in the fixed effects and linear slope in the random effects to 20.2 hours for the model with a cubic spline in the fixed effects and in the random effects. In contrast, the models took 0.03 minutes and 5.05 hours, respectively in the NFBC1966 females where there were 3,280 individuals with 52,162 observations. Therefore, the computational burden may be too large once the sample size gets into the hundreds of thousands. Additionally, large biobank studies, such as the UK Biobank, only have a few repeated measurements and therefore it would be difficult to model non-linear trajectories. The methods developed for phenotypes that change linearly over time may be more appropriate[14–18], or a simple rate of trait change could be derived[45]. However, we recommend these methods be tested within the age groups available in the biobank studies as almost all phenotypes follow non-linear patterns during the life-course[46]. Fourth, we have analysed BMI on the natural log scale for its statistical properties, but this makes the interpretation of effects on BMI more difficult due to the multiplicative errors. Fifth, one of the advantages of using repeated measurements per individual is to increase the statistical power to detect genetic associations when the SNP is included in the fixed effects part of the LMM due to the smaller residual error variance. However, given we are reducing the dimension of our dataset back to a single measure per person (for each GWAS analysis), it is unclear what impact this has on the statistical power. Sixth, we have removed related individuals from the majority of the cohorts (all except CHOP where relatively

little cryptic relatedness was present). However, a random effect for family membership could be included in the LMM if there is a substantial number of relative pairs and family information is available. Finally, BMI as a measure of adiposity during infancy is not commonly used clinically, with ponderal index (weight divided by height[3]) being preferred. However, to model the trajectory from two weeks to 18 years of age we needed to use one measure consistently. Further research into the effect of different powers of height, as investigated by Stergiakouli and colleagues[47], would be of interest.

In conclusion, we have described a framework for conducting GWAS meta-analyses on longitudinal (repeated measures) phenotypes that have a non-linear trajectory over time. We provide an R package, EGGLA, to conduct these analyses for childhood BMI consistently across different cohorts. We have shown that the estimated phenotypes summarise the BMI growth trajectory across a range of cohorts and are able to detect genetic associations with known BMI-associated regions of the genome and to detect new genetic associations. Performing similar analyses across a wider range of cohorts with BMI measures across childhood may identify additional loci for change in BMI across early life. Identification of such loci will enable downstream analyses, such as genetic correlation and causal modelling, to investigate relationships between, for example, early life growth and developmental milestones, cognition and later life cardio-metabolic disease.

## Methods
### Ethics
Ethical approval for the Avon Longitudinal Study of Parents and Children (ALSPAC) study was obtained by the ALSPAC Ethics and Law Committee and local research ethics committees. Ethical approval for NFBC1966 and NFBC1986 was granted by the Northern Ostrobothnia Hospital District Ethical Committee 94/2011 (12.12.2011) and 108/2017 (15.1.2018) respectively in accordance with the declaration of Helsinki. The Research Ethics Board of CHOP approved the study. The study protocols for OBE were approved by local ethics committees. Participants of all studies provided written informed consent.

### Overview
The Early Growth Genetics (EGG) Longitudinal Analysis (EGGLA) framework comprises four main components: application of a LMM to longitudinal data (in this case, BMI measurements between 2 weeks and 18 years of age), model diagnostics, model refinement, and finally GWAS, all provided through a series of functions defined within the EGGLA R package[30,31]. The EGGLA R package was developed to provide a unified approach to harmonise analyses within six distinct cohorts. Although the package and accompanying documentation is specific to this longitudinal analysis with childhood BMI, it (and the described methods) serves as an exemplar for other consortium efforts to model other non-linear traits. The EGGLA R package is therefore publicly available on GitHub; it is also available as part of a Docker image, which provides all necessary tools for conducting LMM and GWAS analyses allowing the analyses to be run non-interactively through either Shell Command Language, Bourne-Again SHell, or interactively through R v4.2.0 (or greater). Further details of the application of the EGGLA R package to cohorts within the EGG consortium are described below.

### Cohorts and study participants
We chose six cohorts to test our proposed methods, which include prospective general population-based birth cohorts from different generations and geographical locations (ALSPAC[22,23], NFBC1966[25] and NFBC1986[26]), a retrospective cohort of children with obesity (OBE[27,28]), and different ethnicities (individuals of African American and European American descent in CHOP[24]).

For the growth modelling, we included all individuals with anthropometric data between the ages of two weeks and 18 years, except for OBE. There was very sparse data between 16 and 18 years in

the OBE cohort, so after careful testing we excluded any measures after 16 years. Measures of height and weight from birth to two weeks were excluded to mitigate the effects of the weight drop arising after birth[48]. We excluded individuals who were part of a multiple birth (i.e. twins, triplets). All individuals with available anthropometric data were included in the growth modelling, regardless of whether they had genetic data, to ensure a precise estimate of the average BMI trajectory within each cohort. Cohort-specific covariates were included in the growth modelling; for example, ALSPAC included a binary indicator of measurement source (questionnaire vs. clinic or health visitor measurement) to allow for differential measurement error.

### Data quality control
We applied an automated algorithm developed by Daymont and colleagues[29], which is available as the R package, growthcleanr ([https://cran.r-project.org/web/packages/growthcleanr/index.html](https://cran.r-project.org/web/packages/growthcleanr/index.html)). This package compares each measurement with a weighted moving average of the individual's other measurements to identify biologically implausible values in height and weight. The cleaning function flags potential data errors, including unit-switch errors (e.g. pounds recorded as kgs or height recorded as weight), very extreme values (i.e. z-score > 25), carried forwards (i.e. values identical over time for the same individual), duplicates (i.e. values recorded on the same day), large height absolute differences (i.e. a decrease in height by more than 3 cm in sequential measurements), single measurements and pairs (i.e. individual SD score for height was compared to their SD score for weight and vice-versa), error load (i.e. exclude individuals who have a substantial proportion of values flagged to be excluded), and finally moderate outliers (i.e. found by calculating the exponentially weighted moving averages for moderate and extreme outliers). We excluded any height or weight values that were flagged as potential data errors (between 4.2% and 16% of measures within each cohort; Supplementary Data 2). We have incorporated the growthcleaner quality control into our EGGLA R package.

After applying the cleaning protocol, measurements of height (m) and weight (kg) were used to derive BMI as weight(kg) / height(m)². As BMI is skewed and heteroskedastic, a natural log transformation was applied before analysis.

### Linear mixed modelling
LMM is one commonly used approach for overcoming some of the challenges in modelling longitudinal data. By selecting an appropriate function for age, the average trajectories of an outcome (i.e. average relationship between age and BMI) can be estimated as fixed effects in the LMM, while variation around this average on the individual level can be estimated as random effects[49,50]. Further details on LMM are given in the Supplementary Note 4.

**Application of LMMs to BMI.** We applied LMMs to model the trajectory of BMI, on a natural-log scale, over time from two weeks to 18 years of age (except OBE, which had data until 16 years). We fit three different functions for age to capture the non-linear slope of the $\log_e(\text{BMI})$ trajectory: 1) a cubic slope for age, 2) linear smoothing splines with knot points at key inflection points on the curve, and 3) cubic smoothing splines with knot points between inflection points in the curve. While we acknowledge that these three functions for age are not the only functions that could have been used to model the non-linearity in the trajectory, we chose these as they range from a simplistic function to model the non-linear trajectory (linear smoothing splines) to more complex function (cubic smoothing splines), but other functions may be more appropriate for comparison when investigating other traits. The linear smoothing splines were included as they have the benefit of being able to be used as the estimated slope phenotypes, rather than requiring estimated slopes to be derived separately. The three models are described by the equations (Eqs. 3–5) below for individual $i$ at time point $t$:

*LMM with cubic slope for age:*

$$\log_e(\text{BMI}_{it}) = \beta_0 + \sum_{j=1}^{3} \beta_j \text{Age}_{it}^j + b_{0i} + \sum_{j=1}^{3} b_{ji} \text{Age}_{it}^j + \varepsilon_{it} \quad (3)$$

*LMM with linear smoothing splines:*

$$\log_e(\text{BMI}_{it}) = \beta_0 + \beta_1 \text{Age}_{it} + \sum_{k=1}^{K} \beta_{k+1}(\text{Age}_{it} - \kappa_k)_+ + b_{0i} + b_{1i}\text{Age}_{it}$$
$$+ \sum_{k=1}^{K} b_{(k+1)i}(\text{Age}_{it} - \kappa_k)_+ + \varepsilon_{it} \quad (4)$$

*LMM with cubic smoothing splines:*

$$\log_e(\text{BMI}_{it}) = \beta_0 + \sum_{j=1}^{3} \beta_j \text{Age}_{it}^j + \sum_{k=1}^{K} \beta_{k+3}(\text{Age}_{it} - \kappa_k)_+^3 + b_{0i}$$
$$+ \sum_{j=1}^{3} b_{ji}\text{Age}_{it}^j + \sum_{k=1}^{K} b_{(k+3)i}(\text{Age}_{it} - \kappa_k)_+^3 + \varepsilon_{it} \quad (5)$$

Where the fixed parameters are represented by $\beta_0, \beta_1, \beta_2,...$ and $\kappa_k$ is the $k^{th}$ knot where:

$$(t - \kappa_k)_+ = \begin{cases} 0 & \text{if } t \le \kappa_k \\ t - \kappa_k & \text{if } t > \kappa_k \end{cases} \quad (6)$$

The parameter estimates for the random effects are represented by $b_{0i}, b_{1i}, b_{2i},...,$ and are assumed to be multivariate normally distributed. $\varepsilon_{it}$ are the error terms assumed to be normally distributed and independent of the parameter estimates for the random effects. For the LMM with linear smoothing splines, the placement of knot points was based on the underlying biology of the BMI trajectory, with piecewise slopes for the time period of infancy to the AR (first-knot point at 5.5 years), a slope from the AR to the pubertal period (second-knot point at 11 years), and finally from the pubertal period through adolescence. We attempted fitting a third-knot point at 9 months, but the model failed to converge. For the LMM with cubic smoothing splines, we a priori fit knot points at 2, 8 and 12 years based on previous studies[51]. For the LMMs with the cubic slope and cubic smoothing splines, we also compared the model fit when reducing the degree of the polynomial in the random effects to decrease the complexity of the model and attempt to assist in model convergence. For example, for the cubic slope LMM, we fit random effects that had a quadratic function of age and a linear function of age, in addition to the cubic function of age. A list of these models is given in the Supplementary Data 3.

The three LMMs were conducted on the $\log_e(\text{BMI})$ data within each cohort stratified for sex, and we conducted models with and without specifying a continuous autocorrelation structure of order 1 (CAR(1)) correlation structure.

**Selection of the best model.** Assessment of model fit was appraised using the following indices of model quality and goodness of fit; $R^2$ (conditional and marginal), intraclass correlation coefficient (ICC), Akaike's Information Criterion (AIC), Bayesian Information Criterion (BIC), root mean squared error (RMSE), and residual SD. The selection of the overall best model was based on the most favourable model performance metrics across all cohorts, focusing on AIC and BIC as they penalize model complexity, as well as model convergence and warning and error messages. It is important to note that the 'best fitting model' for each individual cohort was not necessarily chosen, as each cohort was slightly different for which model fit the data best.

The best model was taken forwards for knot placement refinement. We used a systematic approach to refine the best fitting model

further using an incremental series of knot points where knot 1 was placed at 1, 1.5, and 2 years; knot 2 was placed at 6, 7, and 8 years and knot 3 was placed at 12 years for all models. As for the model diagnostics above, indices of model quality and goodness of fit were used to determine our final BMI trajectory model.

## Estimating phenotypes for GWAS from the BMI trajectories

From the BMI trajectories generated by our model, we sought to extract interpretable characteristics that we could subsequently interrogate for genetic associations. We estimated the age and BMI at the AP and AR, two features of the BMI trajectory that respectively mark the transitions from the rise in BMI across infancy to the decline in BMI in early childhood, then the subsequent rise in BMI from early childhood. In addition, we hypothesised that there may be shared and/or distinct genetic contributions to BMI preceding and succeeding these features[39]. To this end we defined time periods of approximately linear change in BMI prior to the AP, between the AP and AR, from the AR to the approximate onset of puberty and through adolescence. In addition to the rate of change (slopes) over these time intervals, we also analysed the area under the BMI curve (AUC), an estimate of the child's cumulative exposure to excessive body weight. We refer to each of these estimated parameters from the BMI trajectories as the estimated phenotypes.

## Deriving the adiposity peak and rebound

We determined both the age (in years) and BMI (kg/m$^2$) at the AP and AR from the BMI trajectory model. For each participant, $\log_e$(BMI) was predicted using the fixed and random coefficients from the models at intervals of 0.01 year. Then, the AP was defined as the first maximal $\log_e$(BMI) occurring between the ages of 0.25 and 10 years of age, and the AR was determined as the first nadir within that time interval. These cut-off points were chosen based on previous evidence of the mean ages of AP and AR in European populations[9]. The age and BMI for these two points was estimated by taking the exponential of predicted $\log_e$(BMI). We ensured that the age at AP is less than the age at AR; when this did not occur, the individual's age and BMI at both AP and AR were set to missing.

## Choosing the time intervals for deriving slopes and AUCs

A series of time periods of approximately linear change in BMI were defined, avoiding the inflection points at the AP and AR. The time intervals to derive these slopes (and AUCs) were determined visually from the population average BMI trajectories in each cohort and by minimising the proportion of individuals with AP and/or AR falling within the defined time periods (Supplementary Data 1). We chose the following time intervals for derivation of the slopes and AUCs:

- 0 to 6 months (referred to as infancy)
- 1 ½ years to 3 ½ years (early childhood)
- 6 ½ years to 10 years (late childhood)
- 12 years to 17 years (adolescence)

Although these time intervals avoided the average age of the AP and AR, there was a small proportion of individuals with AP and/or AR falling within these time intervals.

## Deriving the slopes

Slopes for each individual for each time interval were estimated using (Eq. 7):

$$\text{Slope}_{b-a} = \frac{y_b - y_a}{x_b - x_a} \qquad (7)$$

Where $y_b$ is the predicted value of $\log_e$(BMI) from the LMM at the ending point for the time interval (e.g. 6 months for the infancy slope), $y_a$ is the predicted value of $\log_e$(BMI) from the LMM at the starting point (e.g. 2 weeks for the infancy slope), and $x_a$ and $x_b$ are the ages at the earlier and later time points, respectively.

## Deriving the area under the curve (AUC)

The AUCs for each time interval were estimated by integrating the best fitting model with respect to age. For example, the AUC for the model with cubic smoothing splines from Eq. (5) above would be estimated using Eq. 8:

$$\int_a^b \left( \beta_0 + \sum_{j=1}^{3} \beta_j \text{Age}_{it}^j + \sum_{k=1}^{K} \beta_{k+3}(\text{Age}_{it} - \kappa_k)_+^3 + b_{0i} \right.$$
$$\left. + \sum_{j=1}^{3} b_{ji}\text{Age}_{it}^j + \sum_{k=1}^{K} b_{(k+3)i}(\text{Age}_{it} - \kappa_k)_+^3 + \varepsilon_{it} \right) d(\text{Age}) \qquad (8)$$

Where $a$ is the earlier time point (e.g. 0 years for the infancy AUC) in each time interval and $b$ is the later time point (e.g. 6 months for the infancy AUC).

## Final model checks

Individuals were flagged as being an outlier for any of the estimated phenotypes (slopes, AUCs, age and BMI at AP/AR) based on the interquartile range (IQR)[52], with values outside twice the IQR flagged as outliers (i.e. two times the IQR above the third quartile and below the first quartile). If an individual was flagged as an outlier for any one of the estimated phenotypes, they were excluded from analyses involving any of the estimated phenotypes. We took this conservative approach to excluding outliers, because if an individual was flagged as an outlier for an estimated phenotype, it likely indicates an issue with their whole predicted curve. Per cohort, this excluded the following proportions of individuals from further analysis: 5% in NFBC1986, 6% in NFBC1966, 9% in ALSPAC, 10% CHOP African American, 12% OBE and 16% in CHOP European American sample.

Correlation matrices between the estimated phenotypes were generated to aid in downstream interpretation of GWAS results. We also conducted association analyses between each of the estimated phenotypes and BMI at the end of the trajectory for each cohort (i.e. BMI at age 16 years in OBE and 18 years for all other cohorts). We converted each of the estimated phenotypes and BMI at the end of the trajectory to z-scores by subtracting the mean and dividing by the standard deviation within each cohort so that we could compare across phenotypes. To account for mean differences between males and females, analyses were adjusted for sex, and adjusted $R^2$ were reported.

To check whether our estimated phenotypes were robust to different underlying LMMs, we estimated the phenotypes from two additional LMMs, one fitting the CAR(1) correlation structure (keeping the fixed and random effects the same as our preferred model) and the second fitting a cubic spline in the random effects in addition to the CAR(1) correlation structure (i.e. the best fitting model according to AIC and BIC in a number of the cohorts – see Supplementary Data 3). We performed these two sensitivity analyses in ALSPAC and NFBC1986, where there were no convergence issues in these updated LMMs. After fitting each LMM, we estimated the slopes and AUCs within each time window, in addition to the age and BMI at the AP and AR and compared them to the estimated phenotypes from the preferred model. We also calculated the correlation matrices and compared them to those from the preferred model.

## GWAS of the estimated phenotypes

Genotyping in each of the contributing cohorts was performed using high-density Illumina BeadChip arrays, and data cleaning and quality control (QC) were performed locally for each cohort (see Supplementary Data 1 for details). Imputation for all European cohorts was performed using the reference data from the Haplotype Reference Consortium (HRC) release 1.1[53]. SNP positions were based on National

Center for Biotechnology Information (NCBI) build 37 (hg19), and alleles were labelled on the positive strand of the reference genome. For the CHOP African American subset, imputation was performed using the TOPMED reference data[54] and SNP positions were based on NCBI build 38.

We performed sex-combined GWAS on the 12 estimated phenotypes derived from the BMI trajectory models: the four estimated slopes, the four AUCs, and age and BMI at AP and AR (OBE performed GWAS on 10 estimated phenotypes as they did not estimate the adolescent slope and AUC due to lack of data after 16 years). Ancestry principal components were included in the models as covariates, along with sex (to adjust for mean differences in the estimated phenotypes between males and females) and cohort-specific covariates where appropriate. GWAS was performed using the imputed allelic dosage data under an additive genetic model. For ALSPAC, NFBC1966, NFBC1986, and OBE, a linear model was applied using PLINK 2.0[55]. For CHOP, analyses using a linear mixed model in REGENIE (version 3.2.6) was used to account for the extended family structure[56]. The sample size for GWAS analysis within each cohort was 6907 for ALSPAC, 5445 for the CHOP European subset, 6332 for the CHOP African American subset, 3579 for NFBC1966, 2887 for NFBC1986 and 490 for OBE (further details in Supplementary Data 1).

## Meta-analysis of the estimated phenotypes

Prior to meta-analysis, variants were first filtered at the cohort level for monomorphic or multiallelic SNPs, indels, low sample size (< 20), low minor allele count (≤ 3), large effect estimates (absolute value of beta or SE ≥ 10 units per allele [log BMI units per year for the estimated slopes, log BMI years for the estimated AUCs, years for the estimated age at AP and AR and kg/m² for the estimated BMI at AP and AR]), and poor imputation quality (INFO < 0.4 or R2 < 0.3). Filtering as well as harmonisation of variants across the cohorts and comparisons of allele frequencies was performed using the EasyQC2 (v. 1.1.1.b5) software package[57] We conducted an inverse-variance weighted fixed-effects meta-analysis, combining each of the five European cohorts, for each of the estimated phenotypes using GWAMA v.2.1[58] and performed a test of heterogeneity in the effect sizes. OBE was excluded from the meta-analysis of the adolescent phenotypes as it did not have data after 16 years of age. Additionally, as OBE is an obesity cohort, we conducted a sensitivity meta-analysis excluding OBE from the other meta-analyses to ensure that it was not driving any observed association. After meta-analysis, we excluded variants that were not present in ≥50% cohorts (i.e. 3 for the European cohorts; 2 for the European cohorts in the sensitivity analyses excluding OBE) as well as variants with minor allele frequency (MAF) < 0.005. Results were clumped and annotated using LocusZoom[59] (version 0.14.0). In the African American subset of CHOP, we selected any genome-wide significant variants identified in the European meta-analysis and performed lift-over to build 37 using UCSC Lift Genome Annotations browser tool (https://genome.ucsc.edu/cgi-bin/hgLiftOver).

Using the genome-wide summary statistics from the meta-analysis, we calculated the SNP-based heritability of each of the estimated phenotypes and the genetic correlations between phenotypes using LD score regression[60,61] (version 1.0.1). We also estimated genetic correlations between the estimated phenotypes and cross-sectional BMI at birth, six months, 1.5 years, 3 years, 7 years, 8 years and adulthood using publicly available summary statistics. The specific ages for childhood BMI were selected so that they would be as close as possible to the cutoff points of our time windows of infancy, early and late childhood. The childhood BMI summary statistics from the Norwegian Mother, Father and Child Cohort Study (MoBA) were reported by[33]. Summary statistics for adult BMI were obtained from the largest meta-analysis to-date of GWAS on cross-sectional adult BMI in individuals of European ancestry[34]. The summary statistics were formatted using a modified version of the supplied munge.sumstats python script. Variants with minor allele frequency less than 1% were removed;

no filtering on imputation quality was performed as this was not available in the meta-analysis. SNP-based heritabilities and genetic correlations were calculated using the European LD scores available from the developers of LD score regression via the genomicSEM website (https://github.com/GenomicSEM/GenomicSEM/wiki).

## Reporting summary

Further information on research design is available in the Nature Portfolio Reporting Summary linked to this article.

## Data availability

To access phenotype and genotype data from individual cohorts participating in the EGG consortium, the cohorts should be contacted directly as each cohort has different data access policies. Full details of each participating cohort can be found in the Supplementary Note 2. Briefly,

ALSPAC: Full instructions for applying for data access can be found here: http://www.bristol.ac.uk/alspac/researchers/access/.

CHOP: CHOP-related data are available upon request from Hakon Hakonarson (hakonarson@chop.edu; response timeframe: one month). Please note that one limitation of the request process is the transfer of data under a material transfer agreement.

NFBC1966: Please, contact the NFBC project center (NFBCprojectcenter(at)oulu.fi) and visit the cohort website (www.oulu.fi/nfbc).

NFBC1986: Please, contact the NFBC project center (NFBCprojectcenter(at)oulu.fi) and visit the cohort website (www.oulu.fi/nfbc).

OBE: OBE-related data are available upon request from Philippe Froguel (p.froguel@imperial.ac.uk; response timeframe: one month). Please note that one limitation of the request process is the transfer of data under a material transfer agreement.

GWAS summary statistics from this study are available via the EGG website (http://egg-consortium.org/longitudinal_growth.html). Source data are provided with this paper.

## Code availability

The analysis pipeline including R code to perform analyses is publicly available from vignettes hosted on https://m.canouil.dev/eggla/articles/eggla.html. The EGGLA R package is available on GitHub: https://github.com/mcanouil/eggla/tree/v1.0.0. See the EGGLA R package record at https://zenodo.org/records/10594717.

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

## Acknowledgements

We would like to thank Associate Professor Loic Yengo for his advice regarding mixed models. We would like to acknowledge the Early Growth Genetics Consortium (http://egg-consortium.org/). A full list of EGG Consortium members appears in Supplementary Note 1. Cohort acknowledgements and funding can be found in the Supplementary Note 2. KB, DAL, and KT contributions to this work are supported by the UK Medical Research Council (MC_UU_00032/02 and MC_UU_00032/05). DAL's contribution is further supported by the British Heart Foundation (CH/F/20/90003 and AA/18/1/34219) and the European Research Council under the European Union's Horizon 2020 research and innovation program (grant agreement No 101021566). AH is supported by the European Union's Horizon 2020 research and innovation program under grant agreement No. 874739 (LongITools), and the Academy of Finland decision 336449 (Profi6). IP and ZB were funded in part by the Diabetes UK (British Diabetic Association no. 20/0006307), European Union's Horizon 2020 research and innovation program (H2020 Science with and for Society) (LONGITOOLS, H2020-SC1-2019-874739). IP is in part is supported by BHF Basic Science Research fellowships (FS/15/59/31839 & FS/SBSRF/21/31025). SFAG is supported by the NICHD (R01 HD056465) and the Daniel B. Burke Endowed Chair for Diabetes Research. SS contributions to this work is supported by grants from the European Commission (LongITools [Grant No. 874739], OBELISK [Grant No. 101080465]; OBCT [Grant No. 101080250]) and the Research Council of Finland (Grant No. 356888). NMW is funded by an Australian National Health and Medical Research Council Investigator grant (APP2008723). The funders of the study had no role in study design, data collection, data analysis, data interpretation, or writing of the report.

## Author contributions

D.A.L., M-R.J., K.T., S.S. and N.M.W. conceived and designed the analyses. M.C. wrote the R package, with contributions from K.B., A.H., J.P.B., M.K. and N.M.W. K.B., A.H., J.P.B., L.N., and M.C. performed the cohort level analyses. A.H, Z.B and M.K. performed the quality control of cohort level GWAS summary statistics and undertook meta-analyses. M.B., Y-M.C., P.F., A.B., H.H., A.C.A., D.A.L., M.K., M-R.J., S.F.A.G., K.T. and I.P., contributed to data quality control, local authority of the birth cohorts, supervision of analyses and/or interpretation of results. I.P., S.S., M.C. and N.M.W. coordinated project activities. K.B., A.H., S.S., M.C. and N.M.W. wrote the manuscript, with participation of all authors.

## Competing interests

DAL received support from Medtronic Ltd and Roche Diagnostics for research unrelated to that presented here. KT acted as Expert Witness to the High Court in England, called by the UK MHRA, defendants in a case on hormonal pregnancy tests and congenital anomalies 2021/22. All other authors report no competing interests.

## Additional information

¹MRC Integrative Epidemiology Unit at the University of Bristol, Bristol, UK. ²Population Health Sciences, Bristol Medical School, University of Bristol, Bristol, UK. ³Research Unit of Population Health, University of Oulu, Oulu, Finland. ⁴Center for Applied Genomics, Children's Hospital of Philadelphia, Philadelphia, PA 19104, USA. ⁵Quantinuum Research LLC, Wayne, PA, USA. ⁶Department of Clinical and Experimental Medicine, School of Biosciences, University of Surrey, Guildford, UK. ⁷People-Centred Artificial Intelligence Institute, University of Surrey, Guildford, UK. ⁸National Heart & Lung Institute, Imperial College London, London, UK. ⁹Univ Lille, INSERM/CNRS UMR1283/8199, EGID, Institut Pasteur de Lille, Lille University Hospital, Lille, France. ¹⁰Division of Endocrinology, Department of Pediatrics, Boston Children's Hospital, Boston, MA, USA. ¹¹Department of Pediatrics, Harvard Medical School, Boston, MA, USA. ¹²Department of Metabolism, Digestion and Reproduction, Imperial College London, London, United Kingdom. ¹³School of Biosciences and Medicine, University of Surrey, Guildford, UK. ¹⁴Department of Epidemiology and Biostatistics, School of Public Health, Imperial College London, London, UK.

[15]MRC Centre for Environment and Health, Department of Epidemiology and Biostatistics, School of Public Health, Imperial College London, London, United Kingdom. [16]Department of Life Sciences, College of Health and Life Sciences, Brunel University London, London, United Kingdom. [17]Center for Spatial and Functional Genomics, Children's Hospital of Philadelphia, Philadelphia, PA 19104, USA. [18]Department of Genetics, Perelman School of Medicine, University of Pennsylvania, Philadelphia, PA 19104, USA. [19]Department of Pediatrics, Perelman School of Medicine, University of Pennsylvania, Philadelphia, PA 19104, USA. [20]Divisions of Human Genetics and Endocrinology and Diabetes, Children's Hospital of Philadelphia, Philadelphia, PA 19104, USA. [21]Institute for Molecular Bioscience, University of Queensland, Brisbane, Australia. [22]Frazer Institute, University of Queensland, Brisbane, Australia. [23]These authors contributed equally: Kimberley Burrows, Anni Heiskala, Jonathan P. Bradfield, Zhanna Balkhiyarova. [24]These authors jointly supervised this work: Inga Prokopenko, Sylvain Sebert, Mickaël Canouil, Nicole M Warrington. ✉e-mail: Sylvain.Sebert@oulu.fi; n.warrington@uq.edu.au

