## [Peer Review File · Nature Communications]

REVIEWER COMMENTS

Reviewer #1 (Remarks to the Author):

Burrows et al., describe in this manuscript a framework to assess the relationship to non-linear longitudinal parameters of any trait with genome-wide data. The manuscript is clearly written and the GitHub created by the authors provides a useful resource for researchers investigating growth curves or age-related decline which in most cases do not follow a linear pattern. The analyses of these trajectories at present are difficult and current methodologies do not address the non-linearity of trajectories. Although I praise the authors for this effort and contributions to the field, I think the wording of some sections could be clearer, and adding some extra information or being more specific will increase the readability of the article.

MAIN REVISION

Title

1. I think the authors need to choose a title that better reflects the work presented in the paper. The title reads “six multiethnic cohorts”. Nevertheless, to my understanding, the only multiethnic cohort is CHOP, and it was analyzed as 2 different cohorts (adding up individually to the six cohorts named in the title). Although the description of the OBE cohort in the supplementary material does not clearly state the ancestral background of the study population, the ethnicity variable reads “European”. Nonetheless, no exclusions based on ethnicity are stated. I would then advise the authors against using multiethnic cohorts to describe the participant studies.
2. I also consider the adjective “time-varying” could be misleading as even if the trait modeling follows trajectories, the framework, by the way pretty original, obtains static parameters derived from these trajectories to follow in the GWAS approach.

Description of the population-specific analyses

3. Through the methods I found it difficult to discern the specificities of the two-step algorithm. Meaning which adjustments were used in the determination of the phenotypes and which were part of the GWAS linear regression. I will provide specific examples.
 - 3.1. In lines 730 to 733, the authors describe the final steps of the quantification of the phenotypes, it reads “We converted each of the estimated phenotypes and BMI at the end of the trajectory to z-scores by subtracting the mean and dividing by the standard deviation within each cohort so that we could compare across phenotypes. Analyses were adjusted for sex, and adjusted R2 were

reported”. Does this mean that the z-scores were sex-specific? Later in lines 745-753, the authors describe the GWAS model, in which each of the generated phenotypes was adjusted for sex and genomic principal components. Does this mean the phenotypes were adjusted by sex and later the GWAS was also adjusted by sex?

3.2. In the description of the cohorts (Supplementary Note), for all cohorts but CHOP exclusion of one participant in case of siblingship is stated. I assume this refers both to the phenotype modeling and GWAS linear regression. Nevertheless, this seems not to have taken place in CHOP, where REGENIE was used for the GWAS step given the complex family structure. How was this structure handled at the phenotype modeling level?

3.3. Also, in the case of the ethnicity definition for CHOP, it is stated in the Supplementary Note that “Self-reported ancestry was used to define African Americans and Europeans Americans”. I assume (but this should be more clearly defined) that this is the definition used to define children taking part of the phenotype modeling. Yet, when the GWAS step takes place, in the sample QC it is stated “Non-African (Non-European) samples were excluded”. As the European part has been dropped off, I assume this is based on an external reference panel (e.g., 1KG or HapMap; This information should be specified). Is this a correct understanding of how ethnicity definition varies in the two steps of the analysis?

4. There are no plots allowing visual assessment of the sparsity of the data across the genomic principal components. Yet, correcting for only 1 PC in the case of CHOP seems low, especially considering the usual diversity in the definition of the “African component”.

5. Although the authors clearly state that the phenotype modeling was based on the entire population and not only on individuals with genotype data, I had trouble finding information about the sample size of each cohort in the GWAS analysis and as such the total sample size of the meta-analysis (maybe I am wrong but I do not think I saw this number stated in any place in the main manuscript, and only in sup table one, per cohort can be found reported).

MINOR REVISION

6. Based on the LDSC paper, it is recommended to report genetic correlations of traits with z-scores of heritability estimates >4 , so that the genetic correlations are interpretable. I guess that’s the threshold used by the authors and which defines the note in Figure 3. I would recommend adding the z-score for heritability to Supplementary table 5.

7. As the authors mention the EGG consortium in the co-author lists, I was expecting to find information on the consortium and affiliated individuals somewhere in the manuscript, but I could not find this information.

Reviewer #1 (Remarks on code availability):

I did not run the code but scan through the structure and help for the user provided

Reviewer #2 (Remarks to the Author):

The manuscript is exceptionally well written, and the Supplementary Information is meticulously organized, demonstrating a commendable level of detail and clarity in the methodological descriptions. However, I do have some concerns regarding the interpretation of the findings and the potential precedent this framework may establish for future research in the field, particularly given its development by distinguished experts.

1. Algorithm convergence issues. While it is understandable that the `lme()` function might encounter convergence issues, relying on software errors as a criterion for model selection may not be advisable. It is evident from the goodness-of-fit measures in Supp Table 3 that models with CAR(1) residuals fit the data best (when `lme()` does not experience convergence problems). Considering the paper's goal to establish a framework within the GWAS field for time-varying traits, it would be advantageous to either propose and implement alternative methods that scale to GWAS without such errors, or to clearly discuss this as an unresolved issue, providing insights into why these convergence issues occur and highlighting the remaining challenges.

2. Opaque temporal correlation structure. The framework utilizes the linear mixed model approach detailed by Pinheiro and Bates (PB; ISBN 1441903178), employing the `nlme` package. PB's model incorporates three components: (i) fixed effects, (ii) random effects, and (iii) autocorrelated or independent residuals within groups. The authors' chosen model incorporates random cubic slope coefficients alongside independent residuals, necessitating that all within-individual residual temporal correlation, conditional on the fixed effects, be modeled by the random coefficients in a polynomial/spline model. However, this approach to modeling temporal autocorrelation may lack transparency and could potentially introduce unintended biases, in particular by introducing artefactual correlations between derived phenotypes (more on this in comment 3 below). These issues might be exacerbated when the origin of the polynomial is set at age zero (relative to, say, average age), and where there is a disparity between the forms of fixed and random effects (spline vs. polynomial). A more transparent, robust and accurate model for within-individual residual variation around fixed effects might involve a smooth stationary time series, such as a stationary CAR(1) or Gaussian process, complemented by white noise.

3. Are derived phenotype results artificially mediated by adolescent/adult BMI? There is a need for a more robust demonstration that the GWAS hits for derived phenotypes are not predominantly

mediated by adolescent or adult BMI, which could merely reflect the residual autocorrelation structure issue described in comment 2, rather than a biological connection. Adjusting the GWAS analyses for a measure of BMI that best represents adolescent/adult BMI in the cohorts would help clarify this. Options include: adjusting for adolescent AUC in other derived phenotypes' GWASs; or adjusting for final BMI, in which case an interaction with age to model systematic trends across different final measurement time points within the 16-18 range might also be beneficially included.

4. Assessment of model fit. The authors provide a wide range of measures of model goodness of fit. BIC and AIC are probably more appropriate than the other measures, as they should hopefully be less likely to overfit the data by penalizing model complexity. Preferable to AIC/BIC, but more computationally intensive, would be cross-validated performance (e.g. cross-validated likelihood). Since they are developing a framework, the authors should carefully discuss the best approaches to model choice, particularly in their current context, where interest is on accurate estimation of within-individual slopes via the predicted random effects.

5. Run times and complexity. A discussion on the computational complexity of the model fitting process, including details on run times and the scalability of the methods for larger datasets, is essential. The datasets used in this study are relatively modest by contemporary biobank standards; thus, demonstrating scalability or acknowledging limitations would be crucial for the framework's application to broader contexts.

Minor comments

6. BMI vs log(BMI). With reference e.g. to Sovis et al 2011 and Warrington 2015, who use log(BMI), the authors should discuss and compare the relative merits of BMI (additive errors) versus log(BMI) (multiplicative errors) in this context.

7. Fixed effects meta analysis. Expanding the main text to include more detail from Supp Table 7 about the tests performed for heterogeneity and the observed minimal evidence of heterogeneity across the European cohorts would be informative.

8. Please clarify the meaning of color used on the diagonal in Figure 3.

Reviewer #3 (Remarks to the Author):

The authors present a new framework/workflow based on non-linear mixed models to fit longitudinal trajectories of traits and to identify variants associated with changes in these trajectories. Using cubic splines, they modeled childhood body mass index (BMI) trajectories from 2 weeks of age to 18 years old. They then computed some summary measures (slope, area under the curve (AUC)) and estimated age at peak adiposity and peak rebound predicted by those models. After running genome-wide association scans (GWAS) on these summary measures, they identified only one novel locus mostly associated with AUC in early childhood, albeit replicating many other loci previously found in adult BMI GWAS.

Fitting the right model and identifying variants related with changes in longitudinal studies is a particularly challenging problem. This novel method is welcomed since most GWAS are using cross-sectional, not longitudinal or time-varying measures. The Methods section gives enough detail to reproduce similar results in other cohorts and an R package has been developed for implementing all the necessary steps before running the GWAS.

Main comments:

- 1) The authors did a great job in fitting the (highly) non-linear childhood BMI trajectory. Unfortunately, when it comes to association testing, one cannot state that this is a true longitudinal or time-varying GWAS, because only summary measures like slope or AUC, i.e. a single measure, is tested for genetic association: the multivariate aspect of it is lost in a sense. Unless some simulations are designed with a variant changing the longitudinal trajectory, it is very difficult to assess if these types of approaches as proposed in this manuscript really increase statistical power. I am not asking authors to compute power estimates, but this would have been a nice addition to the paper.
- 2) In these types of data, we often observe missing values, e.g. due to attrition or missed visits. Do the authors have some kind of rule of thumb with respect to percentages of missing values per individual these types of modeling approaches can tolerate?

3) The cohorts presented in this paper have many observations over time, and most importantly, the measures are somehow observed in the same time window (from birth to 18 years old, except for OBE). However, many electronic health records (EHRs) are now routinely linked to genotype data. In these EHRs, the observations vary a lot with respect to the number of measures, the timing and frequency, even though they are usually restricted to a fixed time window (e.g. up to 10 years before recruitment in a genetic study). As an example, a genetic study could include 80% of participants having 1 or 2 measures for blood pressure, while the remaining 20% have measures every 6 months or every year for 10 years. Could the proposed approach be amenable to this type of scattered observations?

4) Will the authors submit in a near future the eggla R package to CRAN?

Ghislain Rocheleau

Reviewer #3 (Remarks on code availability):

I broadly reviewed the website where one could find all the required steps to reproduce the analysis found in the paper. The eggla R package contains a small dataset in which the cubic splines can be fitted, along with the model selection criteria and diagnostics tools. The package relies on other well-known robust R packages, and on bcftools and plink2.

Reviewer #1 (Remarks to the Author):

Burrows et al., describe in this manuscript a framework to assess the relationship to non-linear longitudinal parameters of any trait with genome-wide data. The manuscript is clearly written and the GitHub created by the authors provides a useful resource for researchers investigating growth curves or age-related decline which in most cases do not follow a linear pattern. The analyses of these trajectories at present are difficult and current methodologies do not address the non-linearity of trajectories. Although I praise the authors for this effort and contributions to the field, I think the wording of some sections could be clearer, and adding some extra information or being more specific will increase the readability of the article.

1. I think the authors need to choose a title that better reflects the work presented in the paper. The title reads “six multiethnic cohorts”. Nevertheless, to my understanding, the only multiethnic cohort is CHOP, and it was analyzed as 2 different cohorts (adding up individually to the six cohorts named in the title). Although the description of the OBE cohort in the supplementary material does not clearly state the ancestral background of the study population, the ethnicity variable reads “European”. Nonetheless, no exclusions based on ethnicity are stated. I would then advise the authors against using multiethnic cohorts to describe the participant studies.

We agree with the reviewer that this may have been misleading, so we have updated the title to “A framework for conducting genome-wide association studies using repeated measures data: An application to body mass index across childhood”

2. I also consider the adjective “time-varying” could be misleading as even if the trait modeling follows trajectories, the framework, by the way pretty original, obtains static parameters derived from these trajectories to follow in the GWAS approach.

We are hesitant to remove all reference to the nature of the trajectory data that we are modelling from our title as this will limit the searchability of the article. However, we have removed ‘time-varying’ from our title and updated it to include ‘using repeated measures data’ to articulate clearer that we are deriving phenotypes from the repeated measures.

3. Through the methods I found it difficult to discern the specificities of the two-step algorithm. Meaning which adjustments were used in the determination of the phenotypes and which were part of the GWAS linear regression. I will provide specific examples.

3.1. In lines 730 to 733, the authors describe the final steps of the quantification of the

phenotypes, it reads “We converted each of the estimated phenotypes and BMI at the end of the trajectory to z-scores by subtracting the mean and dividing by the standard deviation within each cohort so that we could compare across phenotypes. Analyses were adjusted for sex, and adjusted R² were reported”. Does this mean that the z-scores were sex-specific? Later in lines 745-753, the authors describe the GWAS model, in which each of the generated phenotypes was adjusted for sex and genomic principal components. Does this mean the phenotypes were adjusted by sex and later the GWAS was also adjusted by sex?

We apologise for the confusion. Modelling the BMI trajectories was conducted separately in males and females, because we know BMI growth over early life differs by sex. However, this does not account for mean differences in the phenotypes between males and females, so we adjusted for sex in subsequent analyses once we combined the estimated phenotypes from males and females. For the association analysis between final BMI and each of the estimated phenotypes we converted the estimated phenotypes to z-scores – the z-scores were not sex specific which is why we adjusted for sex in the association analyses. In contrast, the GWAS analysis was conducted on the raw estimated phenotypes (i.e., not z-scores) and hence were adjusted for sex and the principal components. The text in the methods section was updated to the following:

“To account for mean differences between males and females, analyses were adjusted for sex, and adjusted R² were reported.”

“Ancestry principal components were included in the models as covariates, along with sex (to adjust for mean differences in the estimated phenotypes between males and females) and cohort-specific covariates where appropriate.”

We also added the following to the results section:

“To select our preferred model, we compared the model fit of sex-specific analyses in each cohort based on any convergence issues, performance metrics and visual inspection of the predicted curves.”

“We calculated a BMI trajectory for each individual within each cohort by combining the estimated fixed effects, which are shared by all subjects within each sex within a cohort, with the predicted random effects, which are specific to each individual.”

3.2. In the description of the cohorts (Supplementary Note), for all cohorts but CHOP exclusion of one participant in case of sibship is stated. I assume this refers both to the phenotype modeling and GWAS linear regression. Nevertheless, this seems not to have taken place in CHOP, where REGENIE was used for the GWAS step given the complex family structure. How was this structure handled at the phenotype modeling

level?

The reviewer is correct that we did not remove related individuals from the CHOP cohort; however, we do not believe this will impact our ability to estimate unbiased genetic effects in the GWAS. Firstly, there is relatively little cryptic relatedness in the cohort when looking at IBD estimates within the genotyped sample. When the GWAS analyses were repeated in the subset of unrelated individuals (determined from the genotype data) using PLINK2 the results were highly concordant with the results on the full genotyped subset using REGENIE. Secondly, the phenotype modelling would ideally include clustering on family structure (i.e., inclusion of a random effect for family membership). However, family information was unavailable and could not be derived on everyone as genotype data was not available on the full cohort. Including relatives in the phenotype modelling and not clustering on family structure would influence estimates of the variances (for example, the standard errors of the fixed effect estimates) but not the means. Therefore, we don't anticipate the estimated phenotypes in CHOP to be substantially affected by the family structure as they are derived from the means.

3.3. Also, in the case of the ethnicity definition for CHOP, it is stated in the Supplementary Note that “Self-reported ancestry was used to define African Americans and Europeans Americans”. I assume (but this should be more clearly defined) that this is the definition used to define children taking part of the phenotype modeling. Yet, when the GWAS step takes place, in the sample QC it is stated “Non-African (Non-European) samples were excluded”. As the European part has been dropped off, I assume this is based on an external reference panel (e.g., 1KG or HapMap; This information should be specified). Is this a correct understanding of how ethnicity definition varies in the two steps of the analysis?

The two subsets in CHOP were defined from the self-report ancestry throughout all analyses presented. However, principal component analysis (PCA) was conducted using the genotype data and any outliers were removed (i.e., those whose PCA results do not align with their self-reported ancestry). The outliers represent a minor fraction of the samples, so an external reference panel, such as 1KG or HapMap, was not used. This is standard procedure for the CHOP cohort and has been previously used in other GWAS publications (for example, <https://pubmed.ncbi.nlm.nih.gov/22484627/>). We have clarified this in the supplementary note:

“Self-reported ancestry was used to define the African American and European American subsets used in all analyses. Self-reported ethnicity was confirmed by principal components analysis.”

And also in Supplementary Table 1:

“Non-European ancestry samples, based on self-reported ancestry, were excluded.”

“Non-African ancestry samples, based on self-reported ancestry, were excluded.”

4. There are no plots allowing visual assessment of the sparsity of the data across the genomic principal components. Yet, correcting for only 1 PC in the case of CHOP seems low, especially considering the usual diversity in the definition of the “African component”.

We have not provided visual assessment of the genomic principal components as these are standard adjustments for GWAS analyses and we have used what is commonly used within each of the cohorts. If the reviewer would like to see a visual assessment, then they can refer to previous publications within each cohort (for example,

<https://pubmed.ncbi.nlm.nih.gov/19060910/>,

<https://www.ncbi.nlm.nih.gov/pmc/articles/PMC6689282/>). In addition, there was an error in the number of genetic principal components that were used in the GWAS analyses in CHOP– we adjusted for 10 principal components to account for minor stratification within the European and African American subsets of CHOP. We have updated this in Supplementary Table 1:

“Sex, first 10 principal components for population stratification”

5. Although the authors clearly state that the phenotype modeling was based on the entire population and not only on individuals with genotype data, I had trouble finding information about the sample size of each cohort in the GWAS analysis and as such the total sample size of the meta-analysis (maybe I am wrong but I do not think I saw this number stated in any place in the main manuscript, and only in sup table one, per cohort can be found reported).

We apologise for not including this in the main text. We have now added this to the Methods Results and Discussion sections.

Methods section:

“The sample size for GWAS analysis within each cohort was 6,907 for ALSPAC, 5,445 for the CHOP European subset, 6,332 for the CHOP African American subset, 3,579 for NFBC1966, 2,887 for NFBC1986 and 490 for OBE (further details in Supplementary Table 1).”

Results section:

“We conducted GWAS for each of the estimated phenotypes within each cohort, then combined the results from the GWAS in cohorts with individuals of European ancestry using fixed effects inverse-variance meta-analysis (combined sample size N=19,308; OBE was

excluded from the meta-analysis of the adolescent phenotypes as they did not have data beyond age 16 years).”

“Twelve of the 28 the genome-wide significant variants identified in the European meta-analysis showed the same direction of effect in the CHOP African American subset (N=6,332) and reached nominal significance ($P < 0.05$) for at least one estimated phenotype (Supplementary Table 7), and a further 12 loci showed the same direction of effect ($P > 0.05$).”

Discussion section:

“We subsequently estimated phenotypes from this LMM and used them to conduct GWAS analyses (N=19,308 individuals of European descent in the meta-analysis and N=6,332 individuals of African American descent).”

6. Based on the LDSC paper, it is recommended to report genetic correlations of traits with z-scores of heritability estimates >4 , so that the genetic correlations are interpretable. I guess that's the threshold used by the authors and which defines the note in Figure 3. I would recommend adding the z-score for heritability to Supplementary table 5.

The reviewer is correct that we used z-score for heritability estimates >4 as our threshold. We have added the z-scores for heritability to Supplementary Table 5 and also updated the Figure 3 title:

“SNP-based heritabilities for AP age, infancy and adolescent slope are low, with high standard errors (resulting in a z-score < 4), and therefore the genetic correlations with these traits are unreliable but are shown for completeness.”

7. As the authors mention the EGG consortium in the co-author lists, I was expecting to find information on the consortium and affiliated individuals somewhere in the manuscript, but I could not find this information.

We have added the EGG consortium members and their affiliations to the supplementary material.

Reviewer #2 (Remarks to the Author):

The manuscript is exceptionally well written, and the Supplementary Information is meticulously organized, demonstrating a commendable level of detail and clarity in the methodological descriptions. However, I do have some concerns regarding the interpretation of the findings and the potential precedent this framework may establish for future research in the field, particularly given its development by distinguished experts.

1. Algorithm convergence issues. While it is understandable that the $\text{lme}()$ function might encounter convergence issues, relying on software errors as a criterion for model selection may not be advisable. It is evident from the goodness-of-fit measures in Supp Table 3 that models with CAR(1) residuals fit the data best (when $\text{lme}()$ does not experience convergence problems). Considering the paper's goal to establish a framework within the GWAS field for time-varying traits, it would be advantageous to either propose and implement alternative methods that scale to GWAS without such errors, or to clearly discuss this as an unresolved issue, providing insights into why these convergence issues occur and highlighting the remaining challenges.

We agree with the reviewer that relying on software errors for model selection is not ideal. However, model convergence issues are a well-known problem when fitting linear mixed effects models to complex data structures like we have (e.g., Bridge *et al.* (2024) Stat Med.) and often indicates that the observed data do not support the need for complex slopes and serially correlated residuals. The most common reason for models not converging is because of a negative estimate of a variance or when the magnitude of the correlation between two random effects reaches or exceeds an absolute value of 1. Unfortunately, it is difficult (impossible) to find out which variance/covariance is causing the issue in most common statistics packages. These convergence issues can often indicate that the model is over-specified, which we suspect is the case when we include the extra parameter when specifying the CAR(1) correlation structure. Along with reducing the complexity of the model specification (as described in the methods section), we also increased the number of iterations the model could take and tested different optimization methods to attempt to get model convergence but with limited success (see the code in the *EGGLA* package for the iterative process we took to try and resolve these issues:

https://github.com/mcanouil/eggla/blob/main/R/egg_model.R). We have added the following paragraph to the discussion outlining some of these issues and how others could attempt to resolve them when applying this framework to their data:

“We acknowledge that we experience a number of convergence issues in the range of models we applied across the cohorts. Although it is not ideal to select models based on whether they converge, it is a practical solution when attempting to apply the same model to a range of datasets with different data structures. Some practical advice to others experiencing convergence issues when attempting to implement our framework include centring the age variable (particularly when zero is not within the age range as the model can struggle to extrapolate to zero), ensuring that each cohort has enough repeated measures of the phenotype (more repeated measurements per individual will allow a more complex random effects structure to be fit), testing different optimization algorithms and

finally, if one of several cohorts is particularly problematic in terms of convergence then investigate the data structure of that cohort and assess whether it is important to include them in the analyses.”

2. Opaque temporal correlation structure. The framework utilizes the linear mixed model approach detailed by Pinheiro and Bates (PB; ISBN 1441903178), employing the nlme package. PB's model incorporates three components: (i) fixed effects, (ii) random effects, and (iii) autocorrelated or independent residuals within groups. The authors' chosen model incorporates random cubic slope coefficients alongside independent residuals, necessitating that all within-individual residual temporal correlation, conditional on the fixed effects, be modeled by the random coefficients in a polynomial/spline model. However, this approach to modeling temporal autocorrelation may lack transparency and could potentially introduce unintended biases, in particular by introducing artefactual correlations between derived phenotypes (more on this in comment 3 below). These issues might be exacerbated when the origin of the polynomial is set at age zero (relative to, say, average age), and where there is a disparity between the forms of fixed and random effects (spline vs. polynomial). A more transparent, robust and accurate model for within-individual residual variation around fixed effects might involve a smooth stationary time series, such as a stationary CAR(1) or Gaussian process, complemented by white noise.

We thank the reviewer for the thought-provoking question. We have conducted two sensitivity analyses to assess the impact of different linear mixed models on our estimated phenotypes:

1. A linear mixed effects model with the cubic spline function for age in the fixed effects, cubic slope function in the random effects and CAR(1) correlation structure for the within-individual residual variation (i.e. the same fixed and random effects as our preferred model, with the addition of the CAR(1) correlation structure).
2. A linear mixed effects model with the cubic spline function for age in the fixed effects and random effects, and CAR(1) correlation structure for the within-individual residual variation (i.e. the best fitting model based on AIC and BIC in the ALSPAC and NFBC1986 cohorts).

We fit these two models in the ALSAPC and NFBC1986 cohorts, where we did not have convergence issues (see Supplementary Table 3, where the error messages are presented for the other cohorts), and then estimated the 12 phenotypes from these modified linear mixed effects models.

The estimated phenotypes were concordant between the preferred model and these two modified models, as seen in the new Supplementary Figures 4 and 5 for the estimated phenotypes in the ALSPAC and NFBC1986 females and males. The slopes and age at adiposity peak and rebound are slightly more sensitive to changes in the underlying model than the AUCs and BMI at the adiposity peak and rebound, which was more evident in the NFBC1986 cohort. Additionally, the correlations between the phenotypes were relatively consistent across the linear mixed models as seen in the new Supplementary Table 5. Therefore, we do not believe that the correlations between the estimated phenotypes are artifacts of the within-individual components fit in the LMM.

We have added the following text to the results section:

“We performed sensitivity analyses in the ALSPAC and NFBC1986 cohorts where we changed the random effects and the correlation structure in the LMM and used these updated models to re-estimate the phenotypes within each time window. The estimated phenotypes were relatively robust to these changes in the underlying LMM (Supplementary Figure 5 and Supplementary Figure 6) and the correlations between the estimated phenotypes were similar (Supplementary Table 5).”

The following to the discussion:

“This indicates that the dependencies between the estimated phenotypes are consistent regardless of the underlying LMM fit to the repeated measures data, which is consistent with our comparison of estimated phenotypes using different structures to model within-individual variation.”

And the methods section:

“To check whether our estimated phenotypes were robust to different underlying LMMs, we estimated the phenotypes from two additional LMMs, one fitting the CAR(1) correlation structure (keeping the fixed and random effects the same as our preferred model) and the second fitting a cubic spline in the random effects in addition to the CAR(1) correlation structure (i.e. the best fitting model according to AIC and BIC in a number of the cohorts – see Supplementary Table 3). We performed these two sensitivity analyses in ALSPAC and NFBC1986, where there were no convergence issues in these updated LMMs. After fitting each LMM, we estimated the slopes and AUCs within each time window, in addition to the age and BMI at the AP and AR and compared them to the estimated phenotypes from the preferred model. We also calculated the correlation matrices and compared them to those from the preferred model.”

3. Are derived phenotype results artificially mediated by adolescent/adult BMI? There is a need for a more robust demonstration that the GWAS hits for derived phenotypes are not predominantly mediated by adolescent or adult BMI, which could merely reflect the residual autocorrelation structure issue described in comment 2, rather than a biological connection. Adjusting the GWAS analyses for a measure of BMI that best represents adolescent/adult BMI in the cohorts would help clarify this. Options include: adjusting for adolescent AUC in other derived phenotypes' GWASs; or adjusting for final BMI, in which case an interaction with age to model systematic trends across different final measurement time points within the 16-18 range might also be beneficially included.

We would expect there to be biological connection between BMI in early life and adult BMI, and we want to use this framework to help understand how genetics contributed to that connection. There have been numerous examples of how genetic loci that influence adult anthropometric traits operate on the phenotype across life (see for example, Hardy *et al* (2010) Human Molecular Genetics, Sovio *et al* (2011) PLOS Genetics, Paternoster *et al* (2011) Human Molecular Genetics, Warrington *et al* (2013) PLOS ONE, Warrington *et al* (2015) Human Molecular Genetics). So, we do not want to adjust out the effect of adult BMI and lose the ability to describe the growth pattern of adult BMI associated loci. Additionally, adolescent/adult BMI cannot mediate the relationship between change in BMI across childhood and genetic variants as it occurs after the time period we are investigating. Therefore, our estimated phenotypes in childhood could mediate the relationship between genetic variants and adolescent/adult BMI, but not the other way around. Finally, by adjusting for a heritable covariate in GWAS analyses, such as adult BMI or adolescent AUC, we could introduce collider bias (Aschard *et al* (2015) American Journal of Human Genetics and Day *et al* (2016) American Journal of Human Genetics) whereby a SNP that is associated with the covariate (here adult BMI or adolescent AUC, whichever is adjusted for) is spuriously associated with the outcome (our estimated phenotypes). Therefore, this would artificially inflate the number of SNPs associated with our estimated phenotypes, rather than adjusting for any residual correlation.

4. Assessment of model fit. The authors provide a wide range of measures of model goodness of fit. BIC and AIC are probably more appropriate than the other measures, as they should hopefully be less likely to overfit the data by penalizing model complexity. Preferable to AIC/BIC, but more computationally intensive, would be cross-validated performance (e.g. cross-validated likelihood). Since they are developing a framework, the authors should carefully discuss the best approaches to

model choice, particularly in their current context, where interest is on accurate estimation of within-individual slopes via the predicted random effects.

We agree with the reviewer that BIC and AIC are the most appropriate criteria for comparing the model fit due to their penalization of overfitting and have updated the manuscript to use these metrics. The other metrics were included to provide insight into model fit within each cohort. Primarily we used BIC for model selection, as it penalizes model fit more than AIC. We have updated Table 2 to show the AIC/BIC metrics and the highlighted cells in Supplementary Table 3 to those models with the lowest AIC and BIC within each cohort/sex. We have also added the following to the methods section:

“The selection of the overall best model was based on the most favourable model performance metrics across all cohorts, **focusing on AIC and BIC as they penalize model complexity**, as well as model convergence and warning and error messages.”

We have also updated the results sections:

“We used **Akaike’s Information Criterion (AIC), Bayesian Information Criterion (BIC)** performance metrics to define the best fitting model **as they appropriately penalize model complexity**. In ALSPAC, **CHOP African Americans**, females in NFBC1966, and NFBC1986 the best model included a cubic spline function for age in both the fixed and random effects and **a CAR(1) correlation structure** (Table 2 and Supplementary Table 3). For the males in NFBC1966 **and OBE**, the best fitting model included a cubic spline function for age in the fixed effects, **a quadratic spline function** in the random effects and **a CAR(1) correlation structure** specified. Finally, the best fitting model in the **male, European American** subset of CHOP had a cubic **spline** in the fixed **effects, a cubic slope** in the random effects, **and no correlation structure and the females was a cubic spline in the fixed effects, linear splines in the random effects and a CAR(1) correlation structure.**”

5. Run times and complexity. A discussion on the computational complexity of the model fitting process, including details on run times and the scalability of the methods for larger datasets, is essential. The datasets used in this study are relatively modest by contemporary biobank standards; thus, demonstrating scalability or acknowledging limitations would be crucial for the framework’s application to broader contexts.

We agree that these datasets are considerably smaller than the large biobanks available; however, they have far more repeated measurements than the biobanks currently. We have added the following to the discussion section of the paper to give readers an understanding of the scalability:

“Third, we have only tested our framework on relatively small cohorts with large numbers of repeated measurements and it is unclear how this will scale to biobank size studies with over one hundred thousand individuals. For instance, in ALSPAC where there are 6,818 samples and 60,169 observations within females, the compute time for each model ranged from 0.04 minutes for the model with a cubic slope in the fixed effects and linear slope in the random effects to 20.2 hours for the model with a cubic spline in the fixed effects and the random effects. In contrast, the models took 0.03 minutes and 5.05 hours respectively in the NFBC1966 females where there were 3,280 individuals with 52,162 observations. Therefore, the computational burden may be too large once the sample size gets into the hundreds of thousands.”

6. BMI vs log(BMI). With reference e.g. to Sovio et al 2011 and Warrington 2015, who use log(BMI), the authors should discuss and compare the relative merits of BMI (additive errors) versus log(BMI) (multiplicative errors) in this context.

We, like Sovio et al. (2011) and Warrington et al. (2015), have used log(BMI) in our analyses. We have added the following to the limitations section of the discussion:

“Fourth, we have analysed BMI on the natural log scale for its statistical properties, but this makes the interpretation of effects on BMI more difficult due to the multiplicative errors.”

7. Fixed effects meta analysis. Expanding the main text to include more detail from Supp Table 7 about the tests performed for heterogeneity and the observed minimal evidence of heterogeneity across the European cohorts would be informative.

We have added the following to the results section of the paper (including two new supplementary figures) to describe the heterogeneity across the fixed effects meta-analysis:

“There was no evidence of heterogeneity across the genome in our fixed effects meta-analysis for the majority of our estimated phenotypes (Supplementary Figure 7). There was some inflation observed for the early childhood slope and the late childhood AUC, but this was driven by the inclusion of the OBE cohort (see Supplementary Figure 8 for Q-Q plots excluding OBE).”

“There was some evidence of heterogeneity ($P < 0.05$) at variants in the *SEC16B*, *ADCY3*, *OLFM4* and *FTO* loci (Supplementary Table 7).”

We also added the following to the methods section:

“We conducted an inverse-variance weighted fixed-effects meta-analysis, combining each of the five European cohorts, for each of the estimated phenotypes using GWAMA v.2.1 (Mägi and Morris 2010) and performed a test of heterogeneity in the effect sizes.”

8. Please clarify the meaning of color used on the diagonal in Figure 3.

We have amended Figure 3 and changed the colour on the diagonal (SNP heritability estimates) to distinguish them from the genetic correlation estimates.

Reviewer #3 (Remarks to the Author):

The authors present a new framework/workflow based on non-linear mixed models to fit longitudinal trajectories of traits and to identify variants associated with changes in these trajectories. Using cubic splines, they modelled childhood body mass index (BMI) trajectories from 2 weeks of age to 18 years old. They then computed some summary measures (slope, area under the curve (AUC)) and estimated age at peak adiposity and peak rebound predicted by those models. After running genome-wide association scans (GWAS) on these summary measures, they identified only one novel locus mostly associated with AUC in early childhood, albeit replicating many other loci previously found in adult BMI GWAS.

Fitting the right model and identifying variants related with changes in longitudinal studies is a particularly challenging problem. This novel method is welcomed since most GWAS are using cross-sectional, not longitudinal or time-varying measures. The Methods section gives enough detail to reproduce similar results in other cohorts and an R package has been developed for implementing all the necessary steps before running the GWAS.

1) The authors did a great job in fitting the (highly) non-linear childhood BMI trajectory. Unfortunately, when it comes to association testing, one cannot state that this a true longitudinal or time-varying GWAS, because only summary measures like slope or AUC, i.e. a single measure, is tested for genetic association: the multivariate aspect of it is lost in a sense. Unless some simulations are designed with a variant changing the longitudinal trajectory, it is very difficult to assess if these types of approaches as proposed in this manuscript really increase statistical power. I am not asking authors to compute power estimates, but this would have been a nice addition to the paper.

We agree with the reviewer that power calculations would be a nice addition, but they were out of the scope of this current study. We do not claim that our two-step approach to modelling repeated measures data increases statistical power over incorporating the genetic data into the LMM, but rather use the two-step approach to reduce computational complexity and increase interpretability of the genetic effect sizes that are estimated.

2) In these types of data, we often observe missing values, e.g. due to attrition or missed visits. Do the authors have some kind of rule of thumb with respect to percentages of missing values per individual these types of modeling approaches can tolerate?

Unfortunately there is no obvious rule of thumb when it comes to rates of missing data. There needs to be enough data so that the model is identified. This will be dependent on how complex the model is that is being fit to the data. For example, our model with the cubic slope in the random effects would require the cohort to have at least four repeated measures per person, but if there is substantial missing data then it is likely that more measures are required. The model assumes that data is missing at random, so the estimated parameters will not be biased by missing data (provided it is missing at random and there is not some underlying structure to the missingness).

3) The cohorts presented in this paper have many observations over time, and most importantly, the measures are somehow observed in the same time window (from birth to 18 years old, except for OBE). However, many electronic health records (EHRs) are now routinely linked to genotype data. In these EHRs, the observations vary a lot with respect to the number of measures, the timing and frequency, even though they are usually restricted to a fixed time window (e.g. up to 10 years before recruitment in a genetic study). As an example, a genetic study could include 80% of participants having 1 or 2 measures for blood pressure, while the remaining 20% have measures every 6 months or every year for 10 years. Could the proposed approach be amenable to this type of scattered observations?

We thank the reviewer for this interesting question. We believe our approach would be amenable to data coming from EHRs in large biobank studies; however, it would be dependent on how many repeated measures are available and what the research question is. Some EHRs only have disease diagnoses over time (for example, using ICD codes) rather than quantitative phenotypes, making these methods more difficult to apply. As mentioned in our response to reviewer 2 question 5, we have not tested our models on large, biobank size datasets, which is a limitation of our current study. Using the example given by the reviewer, where 80% of participants had one or two measures and 20% had very detailed repeated measures, we expect that the 80% of people would get shrinkage to the cohort average effect (as mentioned in the limitations section of the discussion), which would likely have impacts on how well the GWAS could detect novel variants. In this case, a simple rate of change in phenotype, accounting for the difference in age and follow-up time, might be more computationally efficient rather than fitting these highly complex models. A recent paper in Nature Communications has done this using the UK Biobank

(<https://www.nature.com/articles/s41467-024-47802-7>). We have added the following to the discussion:

“Additionally, large biobank studies, such as the UK Biobank, only have a few repeated measurements and therefore it would be difficult to model non-linear trajectories. The methods developed for phenotypes that change linearly over time may be more appropriate (Sikorska et al. 2013; Meirelles et al. 2013; Yuan et al. 2021; Sikorska et al. 2018; Ko et al. 2022), or a simple rate of trait change could be derived (Kemper et al. 2024). However, we recommend these methods be tested within the age groups available in the biobank studies as almost all phenotypes follow non-linear patterns during the life-course.”

4) Will the authors submit in a near future the *eggla* R package to CRAN?

Unfortunately, we are unable to publish the R package to CRAN because of their policies around containing binary programs (PLINK2 here). Unfortunately, PLINK2 versions are not archived, so if we do not incorporate the PLINK2 binary inside our workflow, then we cannot ensure that the same version is used across all cohorts. The R package can easily be installed following instruction from GitHub (<https://github.com/mcanouil/eggla>, <https://m.canouil.dev/eggla/>) or from R Universe (<https://mcanouil.r-universe.dev/eggla>).

REVIEWERS' COMMENTS

Reviewer #1 (Remarks to the Author):

All my points have been addressed properly, for this I thank the authors. Only one remaining point, maybe an small statement regarding how family structure should be addressed within the proposed framework, as a summary of the reply of the authors concerning this point in my previous review.

Reviewer #2 (Remarks to the Author):

The authors have addressed my concerns in this revised manuscript.

Reviewer #2 (Remarks on code availability):

I have not run the code, but the author's website (<https://m.canouil.dev/eggla/articles/eggla.html>) provides detailed instructions for setting up and running their software in a variety of environments.

Reviewer #3 (Remarks to the Author):

I am satisfied with the Authors' responses to my previous comments.

Reviewer #1 (Remarks to the Author):

All my points have been addressed properly, for this I thank the authors. Only one remaining point, maybe an small statement regarding how family structure should be addressed within the proposed framework, as a summary of the reply of the authors concerning this point in my previous review.

We thank the reviewer for their previous comments which were very insightful and helped us to improve our manuscript. We have added the following text to the discussion to address how family structure could be incorporated into our proposed framework:

“Sixth, we have removed related individuals from the majority of the cohorts (all except CHOP where relatively little cryptic relatedness was present). However, a random effect for family membership could be included in the LMM if there is a substantial number of relative pairs and family information is available.”

Reviewer #2 (Remarks to the Author):

The authors have addressed my concerns in this revised manuscript.

Reviewer #2 (Remarks on code availability):

I have not run the code, but the author's website

(<https://m.canouil.dev/egqla/articles/egqla.html>) provides detailed instructions for setting up and running their software in a variety of environments.

We again thank the reviewer for their useful comments on our previous version and are pleased we have appropriately addressed their previous concerns.

Reviewer #3 (Remarks to the Author):

I am satisfied with the Authors' responses to my previous comments.

We would also like to thank reviewer 3 for their previous comments.